# RNA-Based Control of Fungal Pathogens in Plants

**DOI:** 10.3390/ijms241512391

**Published:** 2023-08-03

**Authors:** Christopher W. G. Mann, Anne Sawyer, Donald M. Gardiner, Neena Mitter, Bernard J. Carroll, Andrew L. Eamens

**Affiliations:** 1School of Chemistry and Molecular Biosciences, The University of Queensland, St. Lucia, QLD 4072, Australia; chris.mann@uq.edu.au (C.W.G.M.); a.sawyer@uq.edu.au (A.S.); b.carroll@uq.edu.au (B.J.C.); 2Queensland Alliance for Agriculture and Food Innovation, The University of Queensland, St. Lucia, QLD 4072, Australia; donald.gardiner@uq.edu.au (D.M.G.); n.mitter@uq.edu.au (N.M.); 3School of Health, University of the Sunshine Coast, Maroochydore, QLD 4558, Australia

**Keywords:** gene silencing, RNA interference (RNAi), topical RNAi, cross-kingdom RNAi, spray-induced gene silencing (SIGS), host-induced gene silencing (HIGS), RNA carriers, crop protection, fungal pathogens

## Abstract

Our duty to conserve global natural ecosystems is increasingly in conflict with our need to feed an expanding population. The use of conventional pesticides not only damages the environment and vulnerable biodiversity but can also still fail to prevent crop losses of 20–40% due to pests and pathogens. There is a growing call for more ecologically sustainable pathogen control measures. RNA-based biopesticides offer an eco-friendly alternative to the use of conventional fungicides for crop protection. The genetic modification (GM) of crops remains controversial in many countries, though expression of transgenes inducing pathogen-specific RNA interference (RNAi) has been proven effective against many agronomically important fungal pathogens. The topical application of pathogen-specific RNAi-inducing sprays is a more responsive, GM-free approach to conventional RNAi transgene-based crop protection. The specific targeting of essential pathogen genes, the development of RNAi-nanoparticle carrier spray formulations, and the possible structural modifications to the RNA molecules themselves are crucial to the success of this novel technology. Here, we outline the current understanding of gene silencing pathways in plants and fungi and summarize the pioneering and recent work exploring RNA-based biopesticides for crop protection against fungal pathogens, with a focus on spray-induced gene silencing (SIGS). Further, we discuss factors that could affect the success of RNA-based control strategies, including RNA uptake, stability, amplification, and movement within and between the plant host and pathogen, as well as the cost and design of RNA pesticides.

## 1. Introduction

A surging world population, combined with a changing climate, the increasing vulnerability of natural ecosystems, and the continued use of chemical pesticides, together form a considerable driving force to adopt new and innovative approaches for crop protection [1,2,3,4,5,6]. To feed tomorrow’s hungry populations, new farming practices will need to be developed alongside new tools to protect against pests and pathogens. Currently, the Food and Agriculture Organization of the United Nations (FAO) estimates that yield loss stemming from pests and pathogen infection accounts for approximately 20% to 40% of the potential annual total world crop production. Of these losses, the majority are due to microbial diseases [7], and mycotoxin contamination of foodstuff remains problematic [8]. While conventional fungicides are effective, they are increasingly under scrutiny for their role in causing ecological damage, not to mention developing resistance in emergent pathogen races [9,10]. The use of RNA-based biopesticides as an alternative to the continued use of conventional fungicides is a highly promising example of how biotechnology can develop novel and more environmentally friendly approaches to achieving sustainable food security. In this review, we discuss the current status of RNA-based control of plant fungal pathogens and outline the potential for optimizing this novel and eco-friendly strategy for crop protection.

## 2. RNA-Based Biopesticides

RNA-based biopesticides have been developed around the mechanistic principles of RNA interference (RNAi) and utilize the sequence-specific degradation of complementary transcripts mediated by small regulatory RNAs (sRNAs). In plants, RNAi is synonymous with post-transcriptional gene silencing (PTGS) and functions as a crucial pillar in the defense response against invading viruses, maintains genome stability by silencing transposable elements, and mediates regulation of gene expression during all stages of plant development and in response to biotic and abiotic stress [11]. In fungi, the degree of conservation of gene silencing pathways differs across phyla, but the core protein components of the pathway are retained by most of the agronomically important fungal species [12]. The conservation of gene silencing pathways across most eukaryotes identifies their extreme biological importance, but strikingly, it also enables the possibility of RNA-based crosstalk between species of different kingdoms, so-called cross-kingdom RNAi [13]. A detailed understanding of the core protein machinery and the molecular mechanisms of gene silencing pathways of both the host plant and the invading fungal pathogen will therefore be crucial for the development of effective RNA-based biopesticides as a next-generation crop protection technology. 

### 2.1. Plant Gene Silencing Pathways

In plants, and as detailed in an extensive range of other eukaryotic systems, gene silencing primarily operates at two levels: (1) Transcriptional gene silencing (TGS) mechanisms that direct DNA and chromatin modifications to alter the rate of transcription and (2) post-transcriptional gene silencing (PTGS) mechanisms that degrade homologous RNA molecules and/or suppress their translation [14]. The trigger of gene silencing at either the transcriptional or post-transcriptional level is double-stranded RNA (dsRNA), which can be of endogenous or exogenous origin [14]. Many organisms, including plants, can take up exogenous dsRNA from their environment to varying degrees [15,16,17]. Indeed, this forms the basis for all exogenous RNA-based control measures. Natural sources of dsRNA exist in the form of viruses, which often have single-stranded RNA (ssRNA) genomes that produce dsRNA intermediates as part of their replication. Additionally, virus-specific dsRNAs can be produced from viral ssRNA templates by plant-encoded RNA-dependent RNA polymerases (RDRs) [18,19]. Indeed, it has been repeatedly proposed that the origin and the considerable expansion of gene silencing mechanisms in plants stem from an ancient viral defense mechanism.

Alternatively, the dsRNA trigger for RNA silencing may be derived from the plant genome. In particular, dsRNA can be amplified by plant RDRs from aberrant plant ssRNA templates that lack a messenger RNA (mRNA) cap or poly(A) tail [20,21,22]. Furthermore, many plant species can also be genetically modified to express dsRNAs to direct the silencing of the expression of plant genes, or indeed, the silencing of essential genes in pests and pathogens, in an approach termed host-induced gene silencing (HIGS). 

Within the plant cell, the dsRNA trigger of gene silencing is processed into sRNAs that guide the silencing of complementary nucleic acids [23]. These sRNAs can be very broadly categorized into two subsets based on their production from a specifically structured dsRNA precursor and their primary mode of action. These include: (1) microRNAs (miRNAs) that regulate endogenous-based processes, such as controlling developmental gene expression and gene expression in response to environmental factors; and (2) small-interfering RNAs (siRNAs), which, among other functions, primarily mediate a first line of defense against viral pathogens or the reactivation of transposons [14]. Interestingly, one category of siRNAs is involved in transcriptional gene silencing of transposons and other repetitive DNA elements rather than directing silencing at the post-transcriptional level [24,25]. Crucial to the production of the various sRNA species is the DICER-LIKE (DCL) family of endonucleases, RNase III-like endonucleases that process specifically structured dsRNA substrates into miRNA and siRNA duplexes, typically 20 to 24 nucleotides (nt) in length, with a distinct processing signature of a 2-nt overhang at the 3′ end of each strand of the duplex [26]. Following duplex production, one strand is preferentially selected for loading onto an ARGONAUTE (AGO) protein to form the catalytic core of the RNA-induced silencing complex (RISC), and guided by the mature sRNA, the complex targets complementary nucleic acids for repression at the transcriptional level (chromatin silencing) or the post-transcriptional level (RNA cleavage or translational inhibition) [27,28]. 

Incredibly, plants have evolved the capacity for the silencing signals of RNAi to spread systemically via either the vasculature or a cell-to-cell mechanism of movement [27,28,29]. Crucial to this systemic movement of RNAi in plants is the production of secondary siRNAs by the RDR protein, RDR6 [29]. Both 21-nt and 22-nt sRNAs (miRNAs and siRNAs) can direct AGO1-catalyzed cleavage of target transcripts, but only 22-nt sRNA in complex with AGO1 efficiently recruits RDR6 to initiate amplification of secondary dsRNA using the cleaved target transcripts as templates, even though they are much less abundant than 21-nt siRNAs [29,30,31]. More specifically, RDR6 uses the cleaved ssRNA as a template to synthesize long dsRNAs that are in turn processed by DCL4 and DCL2 to produce abundant 21-nt and low-abundance 22-nt siRNAs, respectively [32]. The role of DCL2-dependent 22-nt siRNAs in the systemic movement of RNAi via the recruitment of RDR6 for secondary siRNA production and for the reception of the systemic silencing signal firmly cements both RDR6 and DCL2 as central RNAi machinery for protection against RNA viruses in plants [29,30]. 

Gene silencing in plants has diversified across evolutionary time to direct an expansive array of biological functions. This is evident in the extensive gene duplication events that have occurred during the evolution of genes that encode core gene silencing machinery proteins and the sometimes overlapping but often distinct functionality of members of these gene families, as detailed below. For example, the model plant *Arabidopsis* encodes four DCLs, ten AGOs, and six RDRs, as well as an extensive suite of other related proteins that together aid in the specification of each RNA silencing pathway [33]. Moreover, sometimes the overlapping functions of members of gene families allow cross-talk to occur between interdependent sRNA pathways that are otherwise involved primarily in mediated silencing mechanisms at either the transcriptional or posttranscriptional level. These RNA silencing pathways include the: (1) miRNA; (2) phased siRNA (phasiRNA); (3) *trans*-acting siRNA (tasiRNA); (4) natural antisense transcript siRNA (natsiRNA); and (5) repeat-associated siRNA (rasiRNA) pathways. 

#### 2.1.1. The miRNA Pathway

Primary miRNA (pri-miRNA) transcripts are transcribed from *MICRORNA* (*MIR*) loci by RNA polymerase II (Pol II) [34,35]. These transcripts are capped, polyadenylated, and, due to partial self-complementarity, fold to form imperfect dsRNA stem-loop structures. The folded region of the pri-miRNA is recognized as a processing substrate by DCL1 together with its functional partners, dsRNA-BINDING1 (DRB1) and SERRATE (SE). Within the nucleus, the DCL1 complex sequentially processes the pri-miRNA into an intermediate stem-loop termed the precursor-miRNA (pre-miRNA) and then into the miRNA/miRNA* duplex (Figure 1) [11]. Each strand of the miRNA/miRNA* duplex is methylated at the 2′ OH group on its 3′ terminal nucleotide by the sRNA-specific methyltransferase, HUA ENHANCER1 (HEN1), a modification that protects the sRNA against degradation [36,37]. It is important to note here that HEN1 also methylates to protect the 3′ ends of all forms of siRNAs in addition to miRNA sRNAs. The mature miRNA guide strand is preferentially loaded into AGO1, while the miRNA* passenger strand is degraded [38]. miRNA-loaded AGO1 forms the catalytic core of miRISC and uses each loaded miRNA as a sequence specificity guide to regulate the expression of mRNA transcripts that harbor highly complementary target sequences [35]. Export of mature miRNA duplexes from the nucleus is facilitated primarily by AGO1, although alternate shuttling routes may be concurrent [39,40]. The human EXPORTIN5 ortholog HASTY (HST), originally thought to be the primary nucleocytoplasmic miRNA exportin protein, has recently been demonstrated to interact with MEDIATOR 37 (MED37) and recruit DCL1 to *MIR* loci [40]. In the cytoplasm, most plant mRNA targets are regulated by AGO1-catalyzed transcript cleavage; however, the inhibition of translation also forms a well-characterized mechanism of miRNA-directed RNA silencing in plants [41,42].

#### 2.1.2. The phasiRNA Pathway

While 21-nt miRNAs primarily function by directing a cleavage or translational inhibition mode of target gene silencing, 22-nt miRNAs also perform another function by directing the biogenesis of secondary siRNAs from *PHAS* loci (Figure 1). These loci encode transcript precursors for phased siRNA (phasiRNA) production, and such *PHAS* transcripts can be either protein-coding RNAs or long non-coding RNAs (lncRNAs) [30,31]. Following cleavage, the now uncapped transcript is converted to a long dsRNA in a RDR6- and SUPPRESSOR OF GENE SILENCING 3 (SGS3)-dependent mechanism [30,31]. In *Arabidopsis* and other dicot species, this dsRNA substrate is processed by DCL4 and DRB4 to cause the iterative production of 21-nt secondary siRNA in a phased manner (head-to-tail processing) [43,44]. The phasiRNAs are then loaded into AGO proteins to direct the cleavage of complementary transcripts [27]. In *Arabidopsis*, *NUCLEOTIDE-BINDING SITE LEUCINE-RICH REPEAT (NBS-LRR)* proteins have been shown to be the most common protein encoding loci from which phasiRNAs are produced, with *PENTATRICOPEPTIDE-REPEAT PROTEIN (PPR)* and *MYB* transcription factor gene family members forming two other common sources for phasiRNA production [45,46]. Interestingly, while relatively few phasiRNAs have been reported in *Arabidopsis*, other plant species are suspected to have many target mRNA transcripts acting as templates for phasiRNA production, especially monocot species, which have a fifth DCL (DCL5) involved specifically in the production of 24-nt phasiRNAs [47,48]. 

#### 2.1.3. The tasiRNA Pathway

Plant *TAS* loci are similar to *PHAS* loci in that a lncRNA is transcribed from each *TAS* locus, which is subsequently used by RDR6/SGS3 as a template for dsRNA synthesis and phased *trans*-acting siRNA (tasiRNA) production. These sRNAs are so named for their silencing of complementary mRNAs encoded by other loci [46,49,50]. Ten distinct *TAS* loci representing four *TAS* gene families (*TAS1–TAS4*) have been well characterized in *Arabidopsis* [51]. As a subset of phasiRNAs, tasiRNAs are derived from cleaved *TAS* transcripts [30]. In *Arabidopsis*, the biogenesis of tasiRNA duplexes is initiated by either 22-nt miRNAs (miR173 and miR828) or a 21-nt miRNA (miR390) that guide AGO1 or AGO7 cleavage of a *TAS* transcript, respectively. The cleavage event recruits RDR6 for dsRNA synthesis [31]. The resultant dsRNA is subsequently processed into phased 21-nt tasiRNA/tasiRNA* duplexes by DCL4 with the assistance of DRB4 [43,50]. Subsequently, specific duplex strands are selected for loading by AGO1 to direct target mRNA silencing [27].

#### 2.1.4. The natsiRNA Pathway

Natural antisense transcripts (*NAT*s) are endogenous RNAs that harbor regions of complementarity with the transcripts of other genes. As initially characterized in *Arabidopsis*, there is an abundance of overlapping natural-antisense gene pairs that, when transcribed from their corresponding DNA template strand, form regions of high complementarity at their paired 3′ ends. The biogenesis of 21-nt natsiRNAs from these highly complementary transcripts involves cross-talk between components of both the TGS and PTGS pathways [21], but the precise steps involved in their biogenesis remain unclear. The biological function of the natsiRNA pathway also remains unclear, but it is upregulated in response to biotic and abiotic stress [21,52], as well as during plant reproduction [53].

#### 2.1.5. The rasiRNA Pathway (RdDM Pathway)

The rasiRNA class of sRNA acts as the silencing signal that directs the highly complex TGS system termed RNA-directed DNA methylation (RdDM) (Figure 1). The canonical RdDM pathway involves the two plant-specific RNA polymerases, Pol IV and Pol V. To date, the rasiRNA-directed RdDM pathway has been associated with the response of a plant to either biotic or abiotic stress, reproductive development, and transposon or transgene silencing [25], and has been speculated to have evolved to enable the maintenance of polyploidy in flowering plant species [24]. DNA methylation is an essential regulatory mechanism in many eukaryotes that is responsible for the epigenetic inactivation of promoters, thereby preventing Pol II transcription [54]. One way that RdDM can be differentiated from the complementary TGS mechanism of histone methylation is the relatively fine resolution of the methylated region: RdDM can target homologous sequences as narrow as 30 base pairs (bp), as compared to histone modification, which has an approximate target resolution of one nucleosome of ~150 bp of DNA in length [55]. 

Pol IV works with RDR2, DCL3, and AGO4 to produce 24-nt rasiRNAs that guide DNA methylation by a suite of factors, including DOMAINS REARRANGED METHYLASE2 (DRM2). Scaffold RNAs transcribed from target loci by Pol V are also required (Figure 1) [24]. A non-canonical RdDM pathway has more recently been identified involving 21- to 22-nt siRNAs derived from the combined activity of Pol II, RDR6, DCL2/DCL4, and AGO4/AGO6. These siRNAs maintain the capacity to also direct mRNA degradation by AGO1-catalyzed cleavage, providing another example of cross-talk between TGS and PTGS pathways [24,56]. In the context of evolutionary significance, following the activation of transposons in response to stress, this cross-talk enables a transition from PTGS to TGS-based silencing of transposable elements [57]. Indeed, the overarching function of the rasiRNA-directed RdDM pathway appears to be the silencing of transposons and other repeat-rich DNA elements to maintain genome stability [58,59].

## 3. Fungal Gene Silencing Pathways

With the exception of a few species such as the corn smut, *Ustilago maydis*, and *Saccharomyces* spp., diverse fungi have been demonstrated to harbor functional gene silencing pathways, and like in plants, fungal gene silencing pathways play a central role in viral defense, the control of transposon movement, developmental regulation of gene expression, stress responses, and to mediate the interactions of fungi with other organisms [60]. However, our current understanding of the molecular complexities of gene silencing in fungi is far from complete. Much of the research conducted to date has been undertaken in the model ascomycetes *Neurospora crassa* and *Shizosaccharomyces pombe*, but gene silencing pathways have been repeatedly reported across the other major fungal groups [12]. Transformation methods have developed slowly for many agronomically important fungi, partly due to their often obligate biotrophic nature, which has hindered a more rapid advancement of our currently limited understanding of the molecular complexities of the fungal gene silencing pathways [61]. That said, the core pieces of protein machinery demonstrated to be essential for gene silencing in plants are conserved in the majority of the fungal species characterized to date, though full characterization of fungal gene silencing remains in its infancy compared to the degree of detailed mechanistic knowledge now available for the plant gene silencing pathways [12,60]. In broad strokes, the canonical core mechanism of gene silencing in fungi can be likened to that of plants, and in all fungal species characterized, it involves DCL-dependent processing of dsRNA triggers into sRNAs, which are in turn loaded into an AGO effector protein to direct a specific mechanism of RNAi or TGS.

Research into gene silencing mechanisms in fungi is further complicated by the complex life cycle of fungal pathogens, and it appears that some components have distinct functionality in specific stages of fungal reproduction. For example, in the widely used model *N. crassa*, a filamentous fungus, RNAi (termed “quelling” in this model system) dominates the asexual stage of development, while DNA methylation and TGS dominate the sexual stage of development [62]. For *Fusarium graminearum*, RNAi machinery is split into sets that contribute unevenly to direct sexual and asexual development as well as pathogenicity [63]. In the agronomically important pathogenic agent of wheat stem rust, *Puccinia graminis* f. sp. *tritici* (*Pgt*), the expression of gene silencing components and the production of sRNA classes differ greatly across the infection timeline, with two distinct waves of sRNA production evident [64]. More specifically, following urediniospore germination and in the early stages of infection, sRNAs were mapped to annotated *Pgt* genes at rates of 77.1% and 68.3%, respectively. However, late wave sRNAs produced 7 days post-infection (dpi) were largely derived from repetitive elements (88.3%), particularly from long terminal repeat (LTR) retrotransposons [64]. Interestingly, these trends were associated with DNA methylation of transposable elements, with late-wave sRNAs more likely to be mapped to methylated transposable elements. This is suggestive of an sRNA-directed DNA methylation pathway, perhaps similar to the rasiRNA-directed RdDM pathway of plants. The absence of stable transformation systems for *Pgt* and for other closely related fungal species makes elucidation of the finer molecular details of such a pathway more challenging than it has been for plants. Despite this hurdle, modest advances toward the identification and/or characterization of other protein machinery components of fungal gene silencing pathways have been achieved to allow for comparison to their plant orthologs.

### 3.1. Core Proteins of Fungal Gene Silencing Pathways

In angiosperms, such as the extensively characterized model species *Arabidopsis*, there is a minimal set of four DCL proteins with well-documented functions. In fungi, DCL protein conservation is more varied, although most have been shown to retain their core function of processing dsRNA substrates into sRNAs [60]. As an example, *F. graminearum*, *Magnaporthe oryzae*, *N. crassa*, and *Botrytis cinerea* have all been shown to encode two DCL proteins [65,66,67], while in contrast, wheat rusts have three *DCL* genes [64]. Interestingly, the conservation of DCL functional domains appears to have deviated between plants and fungi. The PAZ domain, ubiquitous to plant DCLs for its RNA-binding role in determining dsRNA substrate preference, is absent in several notable pathogenic and model fungal species, including *N. crassa* [66], *F. graminearum* [68], *Valsa mali* [69], and *Verticillium nonalfalfae* [70]. In excess of 1000 RNA-binding proteins have been identified in *Arabidopsis* [71], and it seems likely that other eukaryotes, including fungi, would have similarly complex RNA-binding proteomes and that alternative RNA-binding domains could have emerged in these fungal DCL proteins. Indeed, in *V. nonalfalfae* at least, a divergent Dicer/DCL-dimerization domain that also has double-stranded RNA-binding potential has perhaps compensated for the lack of the PAZ domain across both DCLs [70].

In plants, the AGO protein family is more diverse than other components of the gene silencing pathway, with ten distinct AGO (AGO1–AGO10) proteins encoded by *Arabidopsis* and an average of >13 AGOs encoded across all analyzed angiosperm species [72]. As outlined above for the DCLs, the fungal diversification of the AGOs is again more restricted than in plants, although fungal AGOs retain the defining PAZ, MID, and PIWI domains common to all known AGO proteins. For instance, *Parastagonospora nodorum* encodes five AGOs [73], *Pyricularia oryzae* encodes three AGOs [74], *F. graminearum* encodes only two AGOs [68], and *U. maydis* has no AGOs [75] despite their identification in the closely related species, *Ustilago hordei* [76,77]. In *N. crassa*, the AGO protein QUELLING DEFECTIVE-2 (QDE-2) has a well-established role in RNAi [78]. In the plant pathogen *F. graminearum*, *Fg-*DCL2 and *Fg-*AGO1, but not *Fg-*AGO2, are required for RNAi [68]. In *M. oryzae*, which codes for three *Mo-AGO* genes, it was found that the *Mo-ago1* and *Mo-ago3* knockout mutants were deficient in hairpin RNA (hpRNA)- or retrotransposon-induced RNAi [74]. Unexpectedly, *Mo-ago2* knockout mutants exhibited enhanced silencing compared to the wild-type control, a mutation that was also correlated with a reduced growth rate that was only restored upon complementation with the wild-type *Mo-AGO2* gene. These data suggest that under optimal laboratory growth conditions, *Mo-*AGO2 acts to limit *Mo-*AGO1- and/or *Mo-*AGO3-mediated RNAi. Most other agronomically important fungal pathogens have not had the specific functions of their AGO orthologs elucidated to date. Given the complexity and apparent variety of fungal RNAi machinery, thorough characterization of fungal AGOs is a primary priority for future research.

Some fungal pathogens encode their own RDRs that are homologous to plant RDRs [79,80], but secondary siRNA biogenesis in fungi has not been widely reported or well-characterized thus far [80,81,82,83,84]. Indeed, the biochemical function and role of fungal RDRs in RNAi or TGS remain largely unknown. The synthesis of dsRNA by a fungal-encoded RDR was first demonstrated in *N. crassa*, where the RDR, QUELLING DEFECTIVE-1 (QDE-1), was shown to synthesize dsRNA in vivo [85]. The function of QDE-1 in *N. crassa* has since been extensively detailed, and its DNA-dependent RNA Polymerase (DdRP) and RDR functionality in the production of aberrant RNA and dsRNA substrate has been confirmed [78,86,87]. Furthermore, RDR Sad-1 has a confirmed role in the meiotic silencing of unpaired DNA in *N. crassa* [88]. However, most other fungal RDRs remain to be studied in full detail. In *B. cinerea*, three *RDR* genes have been identified bioinformatically [89], but any convincing evidence for their biochemical function remains to be documented. This is mirrored in other agronomically important pathogens, such as wheat rust and *Fusarium* species, where *RDR* genes have been identified but not experimentally characterized. In *Fusarium asiaticum*, however, single knockout mutations of each of its five *RDR* genes failed to result in any mutant line displaying phenotypic variation compared to wild-type *F. asiaticum* [80]. A lack of phenotypic distinction between the *rdr* mutant and wild-type *F. asiaticum* might be accounted for by genetic redundancy, but it is consistent with the finding that secondary siRNA production from the targeted endogenous gene appeared to be lacking in the wild-type pathogen [80]. In *Verticillium dahliae*, the production of secondary siRNAs by fungal RDRs appears to be similarly absent [81], as evidenced by transgenic cotton plants expressing a hpRNA encoding transgene that targeted the *V. dahliae* gene *Hygrophobins1* (*VdH1*), which were resistant to infection, but no accumulation of siRNAs outside the hpRNA-targeted region was found in colonies recovered from infected plants. However, it is entirely possible that RDRs nevertheless play an important role in the silencing of transposons and viruses in *F. asiaticum*, *V. dahliae,* and other fungal species, and additional RNAi pathways involving these proteins are likely to be identified in the future.

Interestingly, in *V. dahliae* colonies that recovered from transgenic cotton plants’ expression of the *VdH1* targeting hpRNA, the expression of *VdH1* could not be fully restored, even though no *VdH1*-specific siRNAs could be detected [81]. One untested possibility to explain these results is that transcriptional downregulation involving DNA methylation or histone modification is responsible for the hpRNA-induced silencing of *VdH1*, and that the DNA methylation or histone modification is subsequently maintained in the absence of the triggering hpRNA. While not widely reported in pathogenic fungi, there have been at least two reports of sRNAs associated with DNA methylation in two fungal pathogens, including *V. dahliae* [64,90], and the existence of a RdDM pathway in fungi has been previously theorized [91]. This poses an interesting possibility for long-lasting pathogen control following transient exposure to a dsRNA with homology to an essential gene in the pathogen.

In summary, detailed knowledge of fungal DCL, AGO, and RDR protein function remains lacking for most agronomically important fungal species compared to what is currently known for their plant host orthologs. More specifically, our knowledge surrounding RDR synthesis of dsRNAs and the subsequent processing of secondary siRNAs from these dsRNAs is severely lacking in fungi. Given the biological importance of secondary siRNA production in plants for viral defense, in addition to the functional importance of RDR activity for transgene silencing [29,92,93], the thorough functional characterization of fungal RDRs, as well as other RNAi pathway components, should remain a research priority. 

### 3.2. The Fungal miRNA Pathway

It was originally thought that fungi lack a functional miRNA pathway [94]. However, strand-specific sRNAs processed from one arm of a stem-loop structured dsRNA precursor transcript provided the first evidence of the existence of a miRNA-like (milRNA) class of small regulatory RNA in fungi. This work, which was performed in *N. crassa*, showed that the identified milRNA sRNAs were categorically distinct from typical miRNAs previously identified in plants due to their highly varied length of 17- to 28-nt, a finding that suggested that either fungi possess a non-canonical miRNA production pathway [94] or that the processing accuracy by DCLs of their dsRNA substrates (i.e., milRNA precursor transcripts) in fungi is reduced compared to that of their plant DCL1 orthologs. The 25 milRNA sRNAs identified were determined to have been processed from only four dsRNA precursor transcripts, termed *pre-milR1* to *pre-milR4*, in an overlapping manner that largely started at the same nucleotide position. Indeed, stem-loop strand and starting nucleotide selectivity strongly suggest that the processing accuracy of milRNA precursor transcripts by fungal DCLs with respect to the 3′ cleavage position is more variable compared to their plant DCL1 orthologs [94].

Following the finding of milRNAs in the model *N. crassa* system, milRNA sRNAs have since been reported in a number of agronomically important fungal pathogens, including *Puccinia triticina* [95], *Puccinia striiformis* [96], *Sclerotinia sclerotiorum* [97], *V. dahliae* [90], and *F. graminearum* [98]. Furthermore, computational methods for predicting milRNA sRNAs and their precursors in fungi have more recently become available [99]. In *S. sclerotiorum,* for example, at least 275 putative milRNAs have been bioinformatically predicted, a prediction that carries a higher degree of confidence, with the differential expression of some of the identified milRNA sRNAs across developmental stages and cell types also correlated to downregulation of their putative targets [97]. In *V. dahliae*, *Vd*milR1 has been shown to play a direct role in target-specific histone modification and epigenetic repression of transcription rather than acting at the post-transcriptional level [90]. It was also shown that *Vd*milR1 sRNA abundance was temporally regulated across the infection timeline, with its target gene *VdHy1* reaching its maximum expression at 3 dpi before downregulation by *Vd*milR1, which reached a maximum level of abundance at 10 dpi. As *VdHy1* expression has been putatively linked to hyphae proliferation, this was theorized to coincide with the requirement to suppress *VdHy1* expression during the biotrophic stage of infection to extend the lifespan of the host plant cells prior to *V. dahliae* transitioning to the saprophytic phase of its growth cycle. In *F. graminearum*, the abundance of *Fg*milR-2 was found to be negatively correlated with the level of expression of its target gene, which encodes the biotin synthesis protein *Fg*bioH1 [98]. Interestingly, *Fg*milR-2 is processed from the 3′ UTR of its *FgbioH1* target transcript, and *Fg*DCL2 was required to produce the *Fg*milR-2 sRNA. These examples show that, similar to the regulatory role of miRNAs in plants and animals, fungal milRNAs also appear to regulate endogenous gene expression to influence specific developmental processes. However, processing of the milRNA precursor transcripts appears to be different in fungi compared to plants, and there is little evidence of miRNA down-regulating transcription in flowering plants.

Interestingly, based on complementarity, milRNAs also have putative targets in their host plants. In the leaf rust pathogen *P. triticina*, three putative fungal *milRNA* precursor transcripts were detected bioinformatically using the ShortStack tool [100] and Vienna RNA folding predictions [101]. One of these milRNAs, *Pt*milR2, was upregulated during infection, which correlated with a downregulation of its putative targets in wheat plants, including a calcium-modulated protein and the CALMODULIN (CAM) and MITOGEN ACTIVATED PROTEIN (MAP) kinases [95]. Similarly, milR1 in the stripe rust pathogen *P. striiformis* has been identified as a potential milRNA-based pathogenicity factor that putatively targets the wheat gene *PATHOGENESIS-RELATED2* (*PR2*) [96]. However, despite these examples, the full scope of cross-kingdom milRNA-directed regulation of host target gene expression remains to be thoroughly substantiated. Indeed, our current collective understanding of the fungal milRNA pathway, including the exact mode of milRNA biosynthesis and the specific regulatory function(s) directed by milRNAs, remains to be fully determined and needs to be elucidated further.

### 3.3. Non-Canonical Fungal Gene Silencing Pathways

Interestingly, in fungi, significant divergence away from canonical gene silencing pathways has been reported. As stated earlier, *N. crassa* clearly possesses milRNA biosynthesis pathway features that differ from the canonical DCL1-directed pathway in plants [94]. Similarly, in *V. dahliae*, the biogenesis of milRNAs appears to be completely different to the canonical DCL processing and AGO loading mechanisms of canonical miRNA pathways, and milRNAs are instead generated from precursor fold-back transcripts by an alternate RNase III domain-containing protein, *Vd*R3 (VDAG_04981). Pol II transcribes *milRNA* loci in *V. dahliae*, whereas Pol III, normally associated with using ribosomal RNA (rRNA), transfer RNA (tRNA) and other sRNA or ncRNA encoding loci as transcription templates, has been reported to be the RNA polymerase responsible for transcription of *milRNA* loci in *N. crassa* [90].

The production of endogenous siRNAs can also differ in fungi from the well-characterized siRNA production pathways in flowering plants. For example, in *Zymoseptoria tritici* mutants defective in DCL activity (*Zt-*Δ*dcl* mutants), siRNAs continue to accumulate to high levels, as well as the *Zt-*Δ*dcl* mutants continuing to display the full virulence and growth characteristics of the wild-type fungus [102]. Further still, in the maize biotroph *U. maydis*, members of both the *DCL* and *AGO* gene families are completely absent, but despite this, sRNAs that are processed in a highly ordered fashion from tRNAs, termed tRNA-derived RNA fragments (tRFs), have been reported to accumulate [75]. These deviations from the canonical sRNA pathways add to the complexity of gene silencing pathways in fungal pathogens and will surely form a considerable body of future research.

## 4. Plant Defense and Cross-Kingdom RNAi

Plant–pathogen interactions have commonly co-evolved to form highly complex species-specific systems. The plant defense response to pathogen invasion has traditionally been explained as a tiered system [103]. The first layer of protection a plant has is its tough and penetration-resistant waxy cuticle that covers the epidermis. However, once this initial barrier has been overcome by the pathogen, whether by mechanical penetration or entry via the stomatal openings, a bidirectional engagement between the microbe and the plant is initiated. Plants detect pathogen-associated molecular patterns (PAMPs), which are recognized by the pattern recognition receptors (PRR) of a plant to initiate PAMP-triggered immunity (PTI), sometimes termed basal immunity [103]. This response induces a chemical alteration to the internal environment of the plant, such as the production of antifungal compounds, as well as a modulation of the regulatory RNA landscape of the plant, such as promoting the production of specific sRNA species [104]. At the same time, the invading pathogen aims to suppress the PTI response of the plant with its own effectors, toxins, and other enzymes. Pathogen effectors are molecules that are trafficked into the plant cell to shut down the plant’s defense response. Unsurprisingly, plant resistance (*R*) genes frequently encode proteins that recognize specific pathogen effectors, which have also been termed avirulence (AVR) proteins [105]. Thus, in natural ecosystems, plant resistance to an invading fungal pathogen is achieved solely by the plant’s natural immune response.

Given the seemingly high conservation of RNAi pathways across plants and fungal pathogens and the complex diversification of pathogen effectors and plant resistance genes, it is not surprising that RNAi has been implicated in host plant–pathogen relations. Not only do endogenous plant sRNAs have a function in regulating the plant defense response upon infection, but recently, plants have been shown to export endogenous sRNAs to knockdown gene targets in their pathogen counterparts. Indeed, this process is believed to be bidirectional, with fungal sRNA effectors also targeting plant genes in a process termed cross-kingdom RNAi. There have been a number of recent studies indicating that sRNA trafficking between host and pathogen plays an important role in pathogen virulence and host resistance [13,65,96,106,107,108,109,110,111,112,113]. On the one hand, fungal sRNAs trafficked into the host plant can function as RNA effectors, shutting down expression of plant defense genes to favor proliferation of the pathogen. Similarly, there is evidence of plant sRNAs being taken into the pathogen to reduce its pathogenicity via sRNA targeting of crucial virulence genes of the pathogen [109,113].

Crop protection derived from HIGS represents one example of cross-kingdom RNAi and involves the uptake of transgene-derived dsRNA and/or sRNA into the pathogen to induce silencing of an essential gene. In the case of endogenous plant sRNA trafficking into fungal pathogens, it has been shown that for cotton (*Gossypium hirsutum*) plants infected by *V. dahliae*, the abundance of two cotton-encoded, highly conserved, and pathogen-targeting miRNAs (miR166 and miR159) is increased upon infection, and these miRNAs are transported into the fungus to downregulate two essential *V. dahliae* virulence genes: one gene encodes a Ca^2+^-dependent cysteine protease (*Clp-1*), and the second gene encodes an isotrichodermin C-15 hydroxylase (*HiC-15*) [113]. Based on these observations, the authors went on to suggest that the capacity for the cotton host plant to mediate miRNA-directed downregulation of fungal targets may have co-evolved in the pathogen to lessen the hypersensitive response and, in doing so, extend the biotrophic stage of infection for the fungus. These examples of cross-kingdom induction of RNAi in the pathogen could therefore be framed as a mechanism to attenuate virulence and thereby ensure the survival of both the host and the infecting pathogen [113]. In another reported example of cross-kingdom RNAi, endogenous *Arabidopsis* tasiRNAs have been implicated in down-regulating virulence genes in infecting *B. cinerea* hyphae [109].

Regarding the transfer of small regulatory RNA from the pathogen to the host plant, it has been reported that *B. cinerea* infection of *Arabidopsis* and tomato is facilitated by the trafficking of transposon-derived sRNAs (*Bc*-sRNAs) of the pathogen into the plant host to shut down the host-encoded immunity genes [65]. Furthermore, the silencing of the same set of *Bc*-sRNA putative target genes in *Arabidopsis* via a HIGS approach showed an increase in the susceptibility of the *Arabidopsis* transformant lines to *Botrytis* infection. In addition, as mentioned earlier in this review, two fungal milRNAs have been shown to act as virulence factors in wheat rust [95,96]. In the apple canker disease pathogen *Valsa mali* (*Vm*), *Vm-*milR1 is complementary to receptor-like kinase mRNAs in its host plant, and *Vm-*milR1 was shown to be highly upregulated during infection [114]. In *Vm*-milR1 deletion mutants, disease symptoms were markedly reduced, with lesion diameters showing an 80–90% reduction compared to those caused by wild-type strains of *V. mali*. Cross-kingdom RNAi induced by pathogen-derived sRNAs is not limited to host plant interactions with fungal pathogens, with sRNAs from the oomycete *Hyaloperonospora arabidopsidis* shown to be transferred and loaded into AGO1 of *Arabidopsis* to direct silencing of a defense gene to enhance the susceptibility of the host to the pathogen [108].

It is interesting to consider whether sRNA effectors may be of equal significance to narrow versus broad host-range pathogens. When considering a pathogen that has co-evolved with a specific host plant species, cross-kingdom RNAi involving fungal sRNAs could be easily factored into an evolutionary arms race model, with the sRNAs functioning as effectors to downregulate resistance mechanisms in the plant host. However, unless a fungal sRNA is relatively abundant and targets highly conserved sequences in plants, it is more difficult to conceptualize how it could have a pivotal role in driving pathogen virulence across a wide range of host species from diverse plant families. In any case, the pathogen sRNAs would represent just one type of effector in a complex array of effector molecules in both narrow and broad host-range pathogens.

The mechanisms behind the bidirectional trafficking of RNA silencing signals between the pathogen and its host have been investigated. In mammalian-parasite interactions, extracellular vesicle (EV) trafficking has an identified role in mediating the transfer of both effectors and nucleic acids [115]. Similarly, EV-mediated protein trafficking between plant hosts and pathogens affects the outcome of host infection by the pathogen [116,117], with evidence suggesting that both host plant and pathogen-derived EVs also play an important role in mediating the transfer of sRNA cargo. This has been primarily explored regarding the delivery of plant-derived sRNAs into *Botrytis* via EV trafficking, resulting in the downregulation of fungal virulence genes [109]. In addition to the *Botrytis*-host plant interaction, there is other evidence that the cargo harbored by plant EVs may impair the virulence of a fungal pathogen. Specifically, sunflower (*Helianthus annuus*) EVs were shown to be taken up by *S. sclerotiorum* and inhibit the virulence of the fungus [118]. Recently, EVs isolated from barley (*Hordeum vulgare*) plants sprayed with a dsRNA construct that targeted the three *F. graminearum CYP51* genes for silencing were found to harbor siRNAs derived from the topically applied RNA [119]. However, when these *CYP51* siRNA-containing EVs were used to treat *F. graminearum* infection, no significant inhibition of the degree of infection or target gene silencing was observed [120].

Several plant proteins have been identified that contribute to the RNA composition of plant EVs and that also affect the outcome of host plant interactions with pathogens [109,121]: findings that are also consistent with EV trafficking playing an important role in the movement of regulatory RNA cargo between the plant host and pathogen. However, on a precautionary note, a recent review of plant EV research has highlighted the possibility of co-isolated cytoplasmic contaminants in crude EV preparations from plant tissues [122], and such contaminants could contribute to the apparent enrichment of some *Arabidopsis*-derived RNAs the EVs detected [109]. Therefore, the RNA composition of EVs and their role in determining the outcome of plant–pathogen interactions need to be investigated further.

## 5. RNA Crop Protection Strategies: HIGS versus SIGS

There are two major RNA-based crop protection strategies: HIGS and spray-induced gene silencing (SIGS). In HIGS, plants are transformed with hairpin RNA (hpRNA) encoding transgenes homologous to pathogen gene targets, conferring protection against the targeted pathogen. This is thought to occur by cross-kingdom trafficking of either the transgene-encoded hpRNA trigger and/or the sRNAs processed in plant cells from the hpRNA. HIGS was first demonstrated to be effective against viral plant pathogens. Indeed, of the 75 studies reviewed by Koch and Wassenegger (2021), it was found that HIGS conferred a mean of 90% resistance against the targeted viruses [123]. More recently, this approach has also been shown to be effective against an array of filamentous pathogens, initially against the obligate biotroph *Blumeria graminis* (powdery mildew) [124], and subsequently against multiple *Fusarium* spp. [125], *B. cinerea* [111], *P. triticina* [126], *Phytophthora infestans* [127], and *S. sclerotiorum* [128]. However, stable transformation protocols for many crop species are unavailable or have otherwise proven problematic. In addition, GM crop varieties that harbor a new gene can take up to 10 years to reach commercial release, and there is an ever-increasing consumer pushback regarding the use of genetically modified organisms (GMOs) in agriculture [129].

An alternative to HIGS that circumvents the limitations of using GM crops is the foliar application of exogenous dsRNAs or siRNAs, typically delivered to the plant in the form of a spray, a technique known as SIGS. Reports of successful protection conferred by SIGS against fungal pathogens are less common than what is currently reported in the literature for the HIGS approach. Nonetheless, notable success using the SIGS approach has been reported against *B. cinerea* [16,111,130,131,132], *F. graminearum* [133], *S. sclerotiorum* [130], *F. asiaticum* [80], and more recently against *Fusarium oxysporum* [134], *Phakopsora pachyrhizi* [135], *Aspergillus niger*, *Rhizoctonia solani* [16], and *Austropuccinia psidii* [79] (Figure 2). The primary drawback of the HIGS approach is the significant challenges associated with stably transforming crop species, including the cost, very long development times, and degree of difficulty in effectively generating genetically modified versions of each crop species. Thus, HIGS cannot respond quickly to crop colonization by a new pathogen. SIGS, on the other hand, is hampered primarily by the short half-life of the applied RNA molecules, including dsRNAs and sRNAs, as well as the limitations related to RNA uptake into both the fungal pathogen and host plant. Solving such challenges is therefore critical for future field applications of the SIGS technology.

## 6. Target Gene Selection for RNA-Based Control of Fungal Pathogens

To deploy RNA-based biopesticides as an efficient crop protection strategy, suitable fungal gene targets must be identified and assessed. RNA-seq data can be used to guide target gene selection, and published transcriptomes are becoming more readily available for this task. The research conducted in this area to date has shown that effective prevention of fungal disease can be achieved by targeting genes important for fungal growth and/or pathogenicity. However, despite several in vitro and in planta screens having now been performed, no consensus currently exists as to which class of target gene provides the optimum degree of pathogen protection to the plant. A consensus is also currently lacking on target region design protocols, though some reports have explored comparing the targeting of different gene regions [80,138]. One of the most extensive target screens using dsRNA treatment was performed on *S. sclerotiorum*, the causal agent of white stem rot in *Brassica napus* (rapeseed). In this work, 20 of the 59 targets screened significantly reduced disease symptoms, whereas the dsRNA targeting of one gene significantly increased the severity of disease symptoms [130]. The targets chosen for this study were selected from a range of pathogenicity-related gene families such as reactive oxygen species response, transcription, and host colonization, but also included essential genes that had previously been identified in the human pathogen *Aspergillus fumigatus* [139], and in that way mirrored the approach to novel conventional fungicide design.

In a recently published work, an array of dsRNA constructs targeting the myrtle rust agent *Austropuccinia psidii* were tested for spray-induced pathogen control [79]. As well as *cytochrome P450 monooxygenase* (*CYP450*), *28S ribosomal RNA* (*28S rRNA*), and three uncharacterized targets upregulated in haustoria, the essential genes *β-tubulin* (*β-TUB*), *translation elongation factor 1-α* (*EF1-α*), *mitogen activated protein kinase* (*MAPK*), *acetyl CoA-transferase* (*ATC*), and *glycine cleavage system-H* (*GCS-H*) were also compared in this work [77]. Of the 11 constructs, eight significantly reduced pustule development in detached leaf assays, with *28S rRNA*, *β-TUB*, and *EF1-α* identified as the three most promising targets. These three targets were shown to significantly inhibit urediniospore germination and the development of infection structures in vitro and were highly effective at preventing infection when sprayed onto 1-year-old *Syzygium jambos* (rose apple) plants. This work [77] suggests that a range of fungal targets may prove to be the most effective strategy to provide protection against a given fungal pathogen, which is promising for future research. Thus, screening for effective fungal gene targets for untested pathogens remains an unavoidable and necessary research priority.

Other studies have focused on the biosynthesis pathway targets of conventional fungicides. Widely used demethylation inhibitor (DMI) fungicides such as tebuconazole, triadimefon, and prochloraz target the ergosterol biosynthesis pathway through inhibition of the cytochrome P450, lanosterol C-14 α-demethylase (CYP51). In pioneering work on *F. graminearum*, the causal agent of the devastating Fusarium head blight (FHB) disease, dramatically reduced virulence was observed when challenging *Arabidopsis* and barley lines that had been molecularly modified to express dsRNA-encoding transgenes that could direct silencing of all three *CYP51* transcript variants of the fungus [125]. Furthermore, the same dsRNAs were demonstrated to effectively confer protection against FHB when topically applied to wheat via spraying, including providing protection against the fungus at tissues distal to the spray contact area [133]. A considerable advantage of targeting the *F. graminearum CYP51* transcript variants by topical application of dsRNA is that they build on existing knowledge of the efficacy of fungicide control and, therefore, allow for a direct comparison of the efficiency of the use of conventional fungicide versus the use of either the HIGS or SIGS approach.

Another target of conventional fungicides such as the benzimidazoles is tubulin production, in particular members of the *β-TUB* gene family [140]. Genes encoding the β-tubulin protein have proven to be effective targets for RNA-based control of a variety of fungal pathogens. Indeed, the antifungal activity of dsRNA complementary to the *F. asiaticum β_2_-tubulin* gene has been tested in vitro and by spray application assays for four pathosystems, including: (1) wheat infected with *F. asiaticum*; (2) barley infected with *M. oryzae*; (3) cucumber infected with *B. cinerea*; and (4) soybean infected with *Colletotrichum truncatum*. In each assessed pathosystem, RNA-mediated knockdown of *β_2_-tubulin* was strongly correlated with (1) reduced disease symptom severity, (2) increased fungicide susceptibility, (3) reduced overall fungal biomass, and (4) impaired and/or abnormal spore germination morphology [138]. Interestingly, the high degree of efficacy reported was achieved even though the target sequence coverage of the dsRNA used was relatively low, with only 10% of the *C. truncatum β_2_-tubulin* transcript potentially being targeted by the sRNAs processed from the dsRNA trigger [138]. These results showcase the promise of the SIGS approach as an effective broad-spectrum antifungal treatment option. However, this work also demonstrates how the targeting of highly conserved sequences and gene regions with SIGS vectors has the potential to cause off-target effects if this is not taken into consideration during vector design. Developing future biopesticides that target highly conserved genes must therefore be approached with a high degree of caution.

A further approach to deducing effective fungal targets of RNAi has been to select genes that are preferentially expressed in appressoria, haustoria, or in the early stages of infection, with the assumption that this will be more effective at reducing fungal growth during the crucially important early stages of infection. Early work investigating gene targets to protect against *P. striiformis* infection suggested that *Barley stripe mosaic virus* (BSMV)-based HIGS potentially directs a higher degree of efficacy when targeting fungal genes preferentially expressed in haustoria rather than those genes that are expressed constitutively [141]. Following this work and with the public availability of genome and transcriptome data for the major *Puccinia* spp., 86 potential *Puccinia graminis* f. sp. *tritici* targets that were preferentially expressed in the haustoria were screened, of which ten showed suppression of disease symptoms and that correlated with reduced target transcript abundance [142]. In the same system, virus-mediated transient silencing of a *Protein kinase A* (*PsCPK1*) subunit, which is dramatically upregulated during the early stages of infection (18 h post inoculation), resulted in an approximate 50% reduction in pustule development [143]. While targets preferentially expressed in haustoria may be effective for HIGS-based crop protection strategies, where siRNAs processed in the plant cells are likely to be taken into the haustorium, the same mechanistic basis may not occur for topically applied dsRNA, as has been reported in myrtle rust [79]. These considerations will be addressed further later in this review.

Some studies have focused on targeting fungal effector genes. In wheat, the powdery mildew resistance locus *Pm3* is rendered ineffective by *SvrPm3^a1/f1^*, a RNase-like effector of the powdery mildew fungus, which compromises fungus recognition by the *Pm3* resistance gene. In wheat plants molecularly modified to express a hpRNA-encoding transgene targeting the RNase-like effector *SvrPm3^a1/f1^*, modest but significant mRNA knockdown was observed, and resistance to powdery mildew was partially restored [107]. Importantly, two other homologous fungal genes encoding RNase-like proteins were also downregulated in proportion to their sequence similarity to *SvrPm3^a1/f1^*. One potential benefit of this approach and others similar to it is that it does not cause the knockdown of essential functions in the pathogen; rather, it enhances the natural resistance of the plant by restoring the function of the resistance gene and the secondary immune response of the plant.

Interestingly, it has also been shown that pathogen effectors can take the form of cross-kingdom RNA signals and that plant resistance can be conferred by silencing these signals as precursors within the pathogen before they reach the host. In wheat, the *PR2* gene encodes a β-1,3-glucanase (EC3.2.1.39) [144], which aids in fungal pathogen defense by hydrolyzing the β-glucan of the fungal cell wall. In *Puccinia striiformis* f. sp. *tritici (Pst)*, the fungal milRNA, *Pst*-milR1, was found to be an important virulence factor mediating the infection of wheat by the fungus, where it directed silencing of *PR2* gene expression [96]. Furthermore, the precursor RNA sequence for the fungal *Pst*-milR1 can be targeted by HIGS to confer resistance against the pathogen. By using a BSMV-based delivery method, a 197-bp hpRNA fragment derived from the fungal precursor of *Pst*-milR1 was introduced into wheat seedlings, and these seedlings were challenged ten days later with *Pst* inoculation. A significant reduction in disease symptom severity was observed at 14 dpi for seedlings treated with the BSMV-*Pst*-milR1 hairpin compared to those inoculated with the empty vector alone [96]. More specifically, reduced disease symptom severity in BSMV-*Pst*-milR1 hairpin-treated seedlings was correlated with 60–70% knockdown of the abundance of the pathogen *Pst*-milR1, a 6–9-fold increase in the expression level of *PR2*, and a significant reduction in *Pst* biomass across the 120-h infection period assessed [96].

Components of gene silencing pathways in fungi have also been identified as potential targets for RNAi-based pathogen control. Weiberg et al. (2013) first identified *B. cinerea DCL*s as potential targets for silencing after discovering *Bc*-sRNA effectors that required *Bc-*DCL activity for their biogenesis [65]. The *Bc-DCL* genes were also chosen as targets based on the reduced mycelial growth of *dcl1* and *dcl2* single and *dcl1* and *dcl2* double mutants. The authors went on to develop a HIGS protection system based on *Arabidopsis* and tomato (*Solanum lycopersicum*) transformant lines expressing a hpRNA targeting both *Bc*-*DCL1* and *Bc*-*DCL2.* The protection offered by this HIGS approach against *B. cinerea* infection was significant. In *Arabidopsis*, an approximate 60% knockdown of the level of expression of both targeted *Bc*-*DCL*s (as measured by RT-qPCR) was correlated with a greater than 70% reduction in relative lesion size and an approximate reduction of 90% in fungal biomass. The impact of this HIGS approach was even more evident in the tomato transformant lines, with no visible lesions and an almost complete (i.e., greater than 99%) reduction in fungal biomass. The authors also reported similar results in *Arabidopsis* when they used HIGS to target the *V. dahliae DCL* genes. Furthermore, the authors additionally showed that topically applied dsRNA and sRNAs targeting the two *Bc-DCL*s directed a similar degree of silencing efficiency to that documented for their HIGS work [65].

Targeting *DCL* genes for knockdown with exogenously applied dsRNA may seem counterintuitive, as the target protein is presumably required for processing the exogenous dsRNA into sRNAs within the fungus. Nevertheless, a number of subsequent publications involving multiple fungal pathogens have since confirmed the successful protection conferred by targeting the *DCL* genes of each pathogen [111,131,132,136]. Islam et al. (2021) demonstrated reduced *B. cinerea* growth in vitro and reduced pathogen lesion area in strawberry (*Fragaria x ananassa*) by dsRNA targeting the *Bc*-*DCL* genes [132]. Similarly, exogenously applied dsRNA targeting the *F. graminearum DCL* and *AGO* genes decreased the expression of the targeted genes, which correlated strongly with an approximate 50% reduction in the severity of disease symptoms in barley [136]. One possible explanation for the effectiveness of this approach is that exogenous dsRNA acts directly on transcriptional downregulation of the target genes rather than acting via DCL processing of the exogenous dsRNA into sRNAs to direct an RNAi mechanism, even though the widely held assumption is that exogenously applied dsRNA works via an RNAi mechanism [65,110].

Some recent studies have questioned the role of fungal DCL proteins in pathogenicity. One of these studies generated *B. cinerea dcl1* and *dcl2* double mutants that were confirmed to be compromised in sRNA biogenesis but retained their full virulence on tomato, *Arabidopsis*, *Nicotiana benthamiana*, and common bean (*Phaseolus vulgaris*) [145]. Furthermore, another group has reported that knockout of both *F. graminearum* AGOs or DCLs had no impact on mycelial growth or virulence when infecting wheat and tomato, while simultaneously identifying that *Fg*-AGO1 and *Fg*-DCL2 were crucial in hpRNA-induced silencing of a polyketide synthase gene (*Fg*PKS12) and *Fg*CYP51A [68]. This is in contradiction to a subsequent study that identified *Fg-DCLs* as potent SIGS targets to confer protection in barley plants against *F. graminearum* infection [136].

There are several possible explanations for these seemingly inconsistent reports on the role of fungal *DCLs* in pathogenicity (also see Section 4). First, it may be that currently unknown off-targets from the dsRNA targeting fungal *DCL* genes could be influencing virulence. These off-target effects would likely involve DCL-independent gene silencing pathways, whose existence is well known in many fungi, though they remain poorly understood [78,90,94]. Secondly, the incomplete downregulation of the DCL-dependent gene silencing pathways by homologous dsRNA may be more impactful on fungal virulence and growth than the complete knockout of the same pathway in the *dcl1* and *dcl2* double mutants. Thirdly, RNAi has also been shown to play a role in protecting fungi, including *B. cinerea* and *F. graminearum*, against mycoviruses. Variations in the mycovirus infection of the fungal pathogens between different laboratories may help account for the inconsistent findings regarding the role of fungal *DCLs* in pathogenicity [146]. Finally, plant-pathogen interactions are extremely complex, and the outcome of each specific interaction depends on many host resistance genes interacting with many pathogen effectors, only some of which may be DCL-dependent sRNAs.

## 7. Factors Mediating the Efficacy of Spray-Induced Gene Silencing (SIGS)

For the SIGS approach to function as an effective system for controlling fungal pathogens, the topically applied RNA must be taken up by the fungus to direct the silencing of essential genes by the gene silencing pathway(s) of the fungus. However, whether the process involves active uptake or diffusion of RNA into the pathogen remains largely unknown. Moreover, the degree to which dsRNA or its sRNA derivatives are taken up into fungal cells from the plant surface, apoplast, vasculature, and/or specific plant tissues likely varies from one pathosystem to another. While RNA uptake directly into several fungal species has been demonstrated, many pathogens are not directly accessible by the spraying of RNA biopesticides or indeed contact fungicides. Thus, RNA uptake into the host plant followed by movement to the infection site will be crucial for developing effective RNA-based biopesticides against many important fungal pathogens.

### 7.1. RNA Uptake and Movement in Plants

Efficient uptake of RNA into the plant vasculature and other tissues following foliar spraying is likely to be dependent on several factors. Exactly how these factors vary between crop species is still a topic that requires considerable research to inform biopesticide design. RNA uptake into plants has been noted to be mediated by: (1) the persistence and stability of the applied RNA on the surface of the plant; (2) the ‘wettability’ of the tissue to which the RNA has been applied, with higher wettability demonstrated to be beneficial to the uptake of the applied RNA; (3) the structure and composition of the cuticle and wax layers of the sprayed plant tissue; and (4) the density and aperture of stomata [147]. Owing to these factors and the variable pore size in plant cell walls, the uptake and systemic movement of exogenous RNA in plants can be limited. Enhancing RNA uptake could be achieved by using surfactants and/or penetrants to aid with tissue surface retention and cuticle penetration. Existing formulae to assist with systemic fungicide uptake have already been developed [148,149,150], and these surfactants have already been useful in facilitating exogenous RNA uptake into plants [151].

A feature of RNAi in plants is the systemic movement of silencing signals across both short distances, in a cell-to-cell mode via the plasmodesmata, and long distances that most likely involve the plant vasculature. The cell-to-cell movement of siRNAs and miRNAs through the plant symplast has been demonstrated by several studies [152,153,154]. Exogenously applied dsRNA can pass through the epidermis and mesophyll layers into the plant vasculature, where it could then be transported to distal tissues [133,155,156]. However, despite ongoing efforts, a sound mechanistic understanding of how exogenous dsRNAs and their derived sRNAs travel systemically in a plant is still lacking. Moreover, the internalization of exogenously applied dsRNA into plant cells has not been widely demonstrated. In SIGS-based protection against plant viruses, processing and amplification of the applied dsRNA are required by the plant, which implies that some level of internalization is achieved. Most research into SIGS against plant viruses, however, utilizes mechanical inoculation that damages the plant tissues and may thus enable internalization of the dsRNA [157,158,159,160]. The relevance of the internalization of exogenous dsRNA and its processing by plant cells to provide protection against fungal pathogens is yet to be elucidated. As discussed elsewhere in this review, in dsRNA-sprayed barley plants infected with *F. graminearum*, the processing of systemically transported dsRNA was shown to occur primarily in the fungus rather than in plant cells [133]. Experiments with wild-type *Arabidopsis* and *dcl2 dcl3 dcl4* triple mutants defective in processing the transgene-derived dsRNA into sRNAs suggest that dsRNA rather than the resulting sRNAs are responsible for HIGS-based resistance to insect pests [161]. However, similar reports are lacking for fungal pathogens.

In addition to the movement of siRNAs and/or dsRNA associated with the systemic spreading of gene silencing, plants also possess the capacity to amplify dsRNA and siRNA signals. As mentioned earlier, the production of tasiRNAs requires the conversion of *TAS* transcripts into dsRNA by RDR6 following miRNA-directed cleavage, with subsequent processing of the dsRNA resulting in the phased production of tasiRNA sRNAs. Similar to tasiRNA biosynthesis, viral RNAs and aberrant plant RNA transcripts can also serve as templates for plant RDR activity for the production of secondary dsRNAs and siRNAs. As noted above, sRNAs can move from cell to cell, and in this way, the spreading of the silencing signal forms an integral part of RNAi in plants. The degree to which secondary sRNA production in fungi might play a role in RNA-based fungicides is largely unknown. Understanding the extent of secondary sRNA biosynthesis in pathogens will help inform adequate and effective dosage and spraying regimes for SIGS and should therefore form the focus of future research. The majority of existing work suggests that intact dsRNA uptake into the fungus is necessary for efficient RNAi [16,17,133], although sRNAs derived from the plant host may still contribute significantly to the efficacy of gene silencing in fungal pathogens [80,81,162].

Given the above considerations, to facilitate the internalization of topically applied dsRNAs by plant cells, two further considerations are required: first, the overall design of the structure of the dsRNA itself, and second, potential delivery vectors/carrier molecules that could be used to enhance the delivery of the dsRNA or its processed sRNAs from the site of application to the site of pathogen infection. For example, smaller dsRNAs and delivery vectors may more readily facilitate RNA uptake into plant tissues and its subsequent apoplastic or symplastic movement to systemic tissues [163]. In high-pressure spraying experiments targeting the *GFP* reporter gene in *N. benthamiana*, siRNAs, but not the dsRNA trigger, were shown to induce RNAi to silence *GFP* expression [164,165]. More specifically, failure to induce RNAi was observed when either a 139- or 322-bp dsRNA was applied to the leaves [164,165]. In contrast, longer dsRNAs have been suggested to be more stable within the plant apoplast and vasculature, as well as to offer the additional benefit of a wider array of potentially “active” sRNA silencing signals following dsRNA processing within the pathogen [102].

In summary, the uptake and systemic movement of topically applied exogenous dsRNAs and their processed sRNAs in plants is a crucial area of research that requires further investigation. The systemic transport of dsRNA molecules throughout the plant provides obvious benefits when considering crop protection strategies. In the context of protection against fungal pathogens, the extent and benefit of cellular internalization and processing of exogenous dsRNAs within plant cells remains largely unknown, and indeed, apoplastically-located exogenous dsRNA within plant tissues may be more effective in controlling these pathogens [16,80,102,133,137,156].

### 7.2. RNA Delivery and Uptake in Fungal Pathogens

RNA uptake into fungal cells is of critical importance for RNAi-mediated protection following topical application of exogenous dsRNA. Indeed, it may be the most crucial contributing factor in determining the efficacy of SIGS against fungal pathogens that harbor their own RNAi pathways [16]. The uptake of exogenous dsRNA and siRNAs into germinating spores of some fungal species in vitro and *in planta* has been reported [79,80,111,133,166,167], and there is a growing body of evidence in the form of HIGS protection assays demonstrating the uptake of dsRNAs and siRNAs by fungi from intact plants, as noted above. For example, *F. graminearum* has been shown to take up both dsRNA and siRNAs from barley plants, and furthermore, the internalized dsRNA can be processed into sRNAs by the fungus to induce silencing of the targeted *CYP51* genes to reduce virulence [133]. Similarly, endogenous cotton miRNAs derived from the host plant direct RNAi of target genes in *V. dahliae* to reduce the virulence of the fungus [113]. However, there remains the question of whether plants are actively packaging and trafficking RNAs into pathogens or whether the uptake of these molecules by the pathogen is by diffusion or incidental to pathogen-induced damage of the plant tissue.

There are three potential modes of exogenous RNA uptake by fungal plant pathogens (Figure 3) that include: (1) direct uptake from the plant surface, which would require no internal passage through the plant; (2) uptake from the plant apoplastic spaces; and/or (3) uptake from within the plant cell cytoplasm, a process that would presumably require the fungal haustoria. In the latter two modes, the question of passive diffusion versus selective packaging and directed trafficking of RNA molecules has not been fully resolved. Currently, evidence for all three modes of uptake has been reported, though uptake into the haustoria has not been definitively proven [16,79,80,111,119,121,130,133,155]. Certainly, there is species-specific variation in fungal RNA uptake efficiency, including at least one reported case in *Z. tritici*, where it was reported that no uptake occurred in germinating spores or extending hyphae [168]. Regardless, a more detailed pathosystem-specific understanding is required for the future development of the most efficacious RNAi-based biopesticides.

A recent study compared the efficiency of dsRNA uptake to disease symptom severity for several fungal pathogens and the oomycete pathogen *P. infestans* [16]. This study showed that the degree of dsRNA uptake varied across the assessed pathogens and that SIGS protection was only effective at reducing disease symptom severity for those pathogens that readily took up the delivered RNA. Five fungal species, including *B. cinerea*, *S. sclerotiorum*, *R. solani*, *A. niger,* and *V. dahliae,* readily took up the dsRNA, whereas dsRNA uptake was barely detectable and failed to be detected, respectively, in the non-pathogenic fungus *Trichoderma virens* and the fungal pathogen *Colletotrichum gloeosporioides*. Exogenous dsRNA uptake was also limited in the oomycete *P. infestans* and depended on cell type. Importantly, this study only tested protection at the site of dsRNA application, with no penetrant included to facilitate dsRNA uptake into the plant prior to pathogen challenge. Thus, dsRNA uptake, when it occurred in a pathogen, may have only been from the plant surface.

Effective protection against the agronomically important fungal pathogen *F. asiaticum* seems to be dependent on exogenous dsRNA uptake into the host plant apoplast or cytoplasm prior to inoculation of the pathogen [80]. In experiments where wheat coleoptiles were treated with fluorescein-tagged *Myo5*-dsRNA before inoculation with *F. asiaticum*, a significant reduction in fungal biomass was observed after 24 h of co-cultivation compared to treatments where the fungus was treated with the dsRNA in vitro for only 12 h prior to its inoculation onto plants. *F. asiaticum* was shown to be deficient in detectable secondary siRNA production despite encoding five *RDR* genes, and silencing of the target gene by exogenous dsRNA was only maintained for five hours post-treatment of the fungus cultured in vitro. It therefore sounds reasonable that intact exogenous dsRNAs, observed up to 8 dpi, or primary siRNAs produced in the plant host from the exogenous dsRNA are being taken into the fungus to induce target gene silencing. The apparent improved protection conferred by dsRNA uptake via the plant apoplast or cytoplasm rather than directly into the fungus may have important implications for topically applied RNA-based biopesticides and systemic protection against at least some pathogens [80,133,162].

Other detached leaf studies have indicated that exogenous dsRNA may initially be taken up by the plant into the apoplast and then subsequently transferred intact into the fungus. Some pioneering work describing hpRNA and siRNA delivery by petiole absorption into *Malus domestica* (domestic apple), *Vitis vinifera* (common grape vine), and *N. benthamiana* suggests that the delivered hpRNA is stable and appears to remain unprocessed in the plant apoplast for at least 10-dpi [156]. In detached barley leaf assays assessing the efficacy of SIGS protection against *F. graminearum* infection, dsRNA uptake and target gene knockdown were revealed to extend beyond the region of application to other distal regions of the detached leaves, and the observed knockdown was reliant upon the processing of the dsRNA trigger by the pathogen [133]. This conclusion was drawn from experiments using a *F. graminearum* mutant (*Fg-*∆*dcl-1*) deficient for DCL-1 activity, where no protection to the plant was conferred by topically applied *CYP51*-dsRNA distal to the inoculation site, and no downregulation of any of the three targeted *CYP51* fungal mRNAs was observed in the mutant. This demonstration showed not only that *F. graminearum* could take up dsRNA from its host plant but also that the intact dsRNA formed the trigger of the observed silencing in the pathogen away from the site of dsRNA application. Indeed, *CYP51* knockdown at the distal inoculation site appeared to be enhanced, consistent with the concept that dsRNA uptake into infection hyphae is more effective compared to dsRNA uptake in germ tubes. Importantly, the *Fg-*∆*dcl-1* strain was, however, compromised at the site of dsRNA application, suggesting that exogenous dsRNA uptake and processing into sRNAs in the plant cytoplasm and transfer of the sRNA to the fungus were sufficient to induce protection at the site of dsRNA application. Further analysis of the resulting sRNA-seq data indicated that *CYP51* sRNAs increased significantly upon infection of both local and distal leaf tissue at the site of dsRNA application. However, no evidence was provided to confirm that the sRNA-seq data represented functional sRNAs rather than simply dsRNA degradation products. Nevertheless, given that the *Fg-*∆*dcl-1* mutant was silenced when infecting a dsRNA-sprayed region and that dsRNA transported beyond the site of application was sufficient to induce silencing in the wild-type fungus, the fungus appears capable of taking up both dsRNAs and sRNAs from the host plant.

Fungal uptake of dsRNA or sRNAs directly from infection hyphae inside plant cells has not been conclusively demonstrated. However, there is evidence to suggest HIGS of essential wheat leaf rust genes does not inhibit formation of the pre-haustorial structure in the pathogen [169], and also that genes highly expressed in the haustoria are effective gene targets for HIGS in wheat stripe rust genes [141]. These findings may suggest that, at a minimum, RNA uptake is possible through the haustoria of rust pathogens. For other species, such as *B. cinerea*, there are reports of RNA uptake by hyphae grown in vitro, with the germ tube region closest to the germinated conidia being the primary site of RNA uptake [170]. The uptake of RNA from an exogenous source into fungal pathogens may involve both haustoria and hyphae but is likely to be pathogen specific.

In insects and nematodes, members of a family of transmembrane proteins that facilitate dsRNA and siRNA movement have been uncovered in RNAi defective screens. In particular, the dsRNA transmembrane transporter protein Systemic RNA Interference Defective-1 (*SID-1*) is a mediator for dsRNA movement in these animals [166]. However, like plants, fungal pathogens lack an ortholog of *SID-1*, and intercellular dsRNA and sRNA movement into fungal cells could occur via the process of endocytosis [167]. In *S. sclerotiorum*, gene knockdown directed by exogenously applied dsRNA was shown to be completely abolished by an inhibitor of endocytosis [167]. The degree to which selective trafficking and EV export play a role in this phenomenon remains to be thoroughly researched. There are clearly many unanswered questions regarding exogenous dsRNA uptake and processing in fungal pathogens, and even less is known about these processes in oomycete pathogens.

## 8. Potential Limitations of SIGS-Based Biopesticides

Topical application of dsRNA as a mainstream crop protection strategy has been hindered by a number of obstacles. The cost of producing molecules of dsRNA and siRNA on a large scale has been prohibitive until recently. Indeed, USD 10,000 per gram for the synthesis of these molecules was typical in 2008, when many pioneering SIGS studies were incipient projects [171]. Fortunately, technological advances in this field have already made an incredible impact on the cost of dsRNA production. The cost of in vitro dsRNA synthesis has fallen dramatically to approximately USD 100 per gram [15], while microbial and cell-free production costs have dropped even further to ~USD 2 and less than USD 1, respectively [172]. At a reasonable estimate of 10 g/ha of dsRNA for field applications, this would bring the cost of using dsRNA sprays in line with the use of traditional fungicides [173,174].

Perhaps more importantly, the half-life of RNA is significantly shorter when compared to conventional fungicides. When duplexed into dsRNA, the stability is much improved, but it is still a biological substrate susceptible to enzymatic and physical degradation. However, carriers have been developed to protect against RNA degradation, with considerable success [158]. Accordingly, developing novel approaches to stabilize or protect the RNA without impairing its functionality and activity is an attractive direction for future research. This would aid not only in the field but would also extend the shelf life of the developed formulations. It should be noted that safety testing and regulation of SIGS-based biopesticides are still developing, and data requirements for product registration remain unspecified in Europe, Australia, and the United States [175]. Nevertheless, the impact of RNA biopesticides on non-target species, particularly arthropods, remains a crucial hurdle for product registration as well as a critical area of ongoing research [175,176]. While these potential limitations exist, topical application of RNA for crop protection offers an eco-friendly alternative to traditional fungicides while remaining non-toxic and GM-free [175].

## 9. Potential Limitations of the HIGS Approach

HIGS requires that the crop species be amenable to genetic modification, a process that is time-consuming and costly and that remains unfeasible for many crop species. Despite this, new or modified transformation procedures for recalcitrant crop species are being steadily developed [177,178], and the next generation of transformation protocols seems imminent [179]. The delivery of genetic material for transformation using nanomaterials is one such innovation that can reduce transformation costs and improve reliability [180].

One recurrent issue plaguing HIGS transgenic plant lines is that hpRNA encoding transgenes, which feature inverted repeat (IR) sequences, undergo RdDM and transcriptional silencing of the promoter sequence, which directs the expression of the hpRNA transgene. A recently reported solution to this problem utilizes the replacement of cytosine nucleotides with uracil nucleotides in one strand of the hpRNA. This modification allows guanine–uracil (G:U) “wobble” base pairs to continue to form as well as to ensure that the formation of the hpRNA structure is maintained while the repressive mechanisms of RdDM and transcriptional silencing of the transgene promoter are disrupted [181]. In parallel experiments, transgenic plants expressing G:U hpRNA transgenes outperformed those with unmodified hpRNA transgenes, with effective silencing observed in 90–96% and 57–65% of the respective transformant populations [179]. This enhanced RNAi corresponded to lower levels of RdDM and almost no self-silencing of the promoter driving the expression of the hpRNA coding regions [179]. Innovations such as this are promising solutions to ongoing hurdles associated with the molecular modification of crop species. However, such an approach still fails to address public concerns over GM crop safety and the time needed to produce a new GM crop variety.

## 10. Recent Advances and Prospects for the SIGS Approach

### 10.1. RNA Carriers

One promising approach to extending the half-life of exogenously applied dsRNA is by complexing it with carrier molecules, particularly nanoparticles. As well as protecting the RNA from environmental degradation, the nanoparticles can facilitate cellular internalization of the applied RNA. So far, a range of carriers have been reported, including both carbon and gold nanoparticles [182,183], minicells [132], quantum dots [184,185], and layered double hydroxides (LDH) [134,151,158,186]. For the widespread application of RNA-based biopesticides, carrier formulations must be developed that are cost-effective, degradable, and non-toxic. LDH nanosheets fit these criteria and, when complexed with dsRNA in a formulation known as BioClay^TM^, have been shown to extend the protection window conferred by the exogenously applied dsRNAs against plant viruses, insect pests, and fungal pathogens [131,151,158,159]. These clay nanoparticles, first developed for human therapeutics, are positively charged to enable the negatively charged dsRNA to bind, conferring protection against runoff and nuclease attack while also facilitating sustained release of the dsRNA for long-term protection against the targeted pathogen. The development of such non-toxic RNA carriers is crucial to the future rollout of RNA-based biopesticides.

An additional consideration for nanocarrier-RNA formulations is the efficacy of cellular internalization by the particles and their cargo. Recently, LDH nanosheets with a diameter of up to 40–50 nanometers (nm) were shown to efficiently internalize and deliver exogenous dsRNAs and siRNAs into plant cells to direct RNAi of transgene reporters [163,186]. Similarly, gold nanoparticles of 20 nm diameter have been shown to enable internalization of bound siRNAs [187]. In both cases, the internalization of dsRNAs or siRNAs into tomato pollen or *N. benthamiana* leaves was employed to successfully direct the silencing of a reporter transgene. These small nanoparticles also have the potential to efficiently deliver dsRNA into fungal cells and enhance RNA-based control of fungal pathogens.

### 10.2. Structural Modifications of RNA

The structural modification of RNA to confer enhanced mobility and stability is an under-researched area that is likely to become a central research focus as RNA-based crop protection strategies gain further momentum. At a basic level, careful design of the RNA construct to optimize for length, stability, and mobility is one such approach, but more sophisticated options are emerging [102]. Other modifications could include incorporating sequence motifs for enhanced systemic movement of RNA in plants. Given the ubiquity of RNA-binding proteins (RBPs) in plants—for example, almost ~7% of the *Arabidopsis* proteome is confirmed or putatively associated with RBPs—the conception of “naked” RNA without associated binding proteins is increasingly unlikely [188]. As yet, the prevalence of RBPs in the plant apoplast and vasculature with affinity for long dsRNAs is largely unknown, but it seems likely that, at a minimum, certain dsRNA sequences will be favorable for enhanced translocation, even in the absence of specific dsRNA transporters such as the insect and nematode orthologs of SID-1. RNA interactome capture (RIC) was first employed in plants in 2016, and in the brief span of intervening years since its first report, an amazing depth of RNA-protein interactions has been revealed but not fully characterized [188]. Identifying putative RBPs that are active in the xylem and phloem and specialized for dsRNA- or sRNA-binding may yield critical insights for enhanced movement of topically applied RNA in plants.

In the genomes of plant RNA viruses, the presence of transfer RNA-like structures (TLSs) has been widely reported [189,190,191,192,193,194]. The specific role of these structures is dependent on the virus in question, but stability is one well-established function [195]. A further function is phloem transport. This is supported by the presence of structured RNAs, such as those with TLSs, harbored by both plant and viral RNAs and localized to the phloem [196,197]. It has been shown that the TLSs of *Brome mosaic virus*, *Tobacco mosaic virus,* and *Turnip yellow mosaic virus* could all confer phloem transport of transiently expressed transcripts. In further work in this area, it was shown that over 10% of endogenous mRNAs are graft-transmissible in plants, and many of the graft-transmissible mRNAs contain a TLS [198,199]. Furthermore, including a TLS fused to the 3′ UTR of mRNAs in transformed plants conferred graft-transmissibility to otherwise non-mobile transcripts [198]. Indeed, TLS-conferred transcript mobility has been shown to enhance functional RNA movement across graft junctions, parasitic plant haustoria, and distal tissues [198,200,201]. These findings are consistent with the conception of three-dimensional RNA motifs stabilizing mRNAs and/or being recognized as phloem transport signals (PTSs) by hitherto unknown proteins. Some candidate proteins that recognize PTSs within RNA, such as the *Cucurbita maxima* PHLOEM SERPIN 1 (*Cm*PSI) protein, have been putatively identified, although so far, no RBP that mediates the transport of TLS-containing RNAs has been experimentally validated [202]. Beyond sequence-derived secondary and tertiary structures, these three-dimensional RNA motifs are likely to also feature chemical modifications such as methylation or aminoacylation, which would also change the conformation of the transcript. In support of this, it has been shown that mobile plant mRNAs are also significantly enriched for methyl-5 cytosine (m^5^C) modifications and that this modification also confers mobility over graft junctions [203]. Identifying and incorporating structural motifs that favor translocation through the plant vasculature to enhance the protection conferred by RNA biopesticides is an attractive avenue for future research. If such motifs could also enhance RNA stability and uptake by fungal pathogens, then further improvement of RNA-based pesticides may be possible.

### 10.3. Artificial miRNA (amiRNA)

Over the last two decades, artificial miRNAs (amiRNAs) have been designed as an alternative trigger of RNA silencing to the traditional use of either dsRNA or siRNA triggers in plants. First generated in human cell lines [204], amiRNA-induced RNAi was quickly adopted for use in *Arabidopsis* [205], where it has since been largely used for knockdown of transgenes and endogenous gene targets in plants. This approach uses endogenous primary miRNA or precursor miRNA transcripts as a scaffold for amiRNA delivery via the replacement of the mature sequences of the endogenous miRNA and miRNA* sequences with an amiRNA (and amiRNA*) designed to have complementarity against a desired target gene while maintaining the natural structure of the endogenous precursor transcript. Due to the maintenance of the secondary structure of the precursor transcript, the endogenous protein machinery of the plant miRNA pathway recognizes the amiRNA-containing precursor and processes the precursor into a mature duplex capable of guiding highly specific silencing of the targeted transcript through the usual AGO1-catalyzed mechanisms of miRISC.

As a crop protection strategy, amiRNAs are yet to be properly explored, and only a limited number of studies have employed amiRNAs in the HIGS system. Thus far, amiRNA-based HIGS methods have focused largely on viral and insect pests, and only a handful of studies have targeted other classes of pathogens. Against the oomycete *P. infestans*, some modest protection was observed in potato (*Solanum tuberosum*) plants transformed to express an amiRNA targeting the *P. infestans* virulence effector *Avr3a*. The protection offered by this amiRNA-based HIGS approach correlated with rapid downregulation of *Avr3a* transcript abundance in infected tissue extracted from partially resistant transformed plants [206]. In another example, amiRNAs expressed in molecularly modified wheat plants that mimicked the sequence composition of virus-derived siRNAs were shown to confer broad resistance to several viral pathogens as well as *P. striiformis*. The broad pathogen resistance observed was subsequently shown to be the result of the stimulation of basal immunity caused by amiRNA-directed inhibition of the ROS scavenging pathway [207]. However, a HIGS strategy based on an amiRNA targeting an essential gene of a fungal pathogen has yet to be reported.

Interestingly, the instances of using amiRNAs to induce RNAi in plants have all used plant miRNA precursors as backbones. With more recent studies confirming the activity of fungal milRNAs, the opportunity exists to test amilRNA-directed RNAi in fungal pathogens using milRNA precursors as the delivery transcripts in either a HIGS or SIGS approach. How these fungal amilRNA will behave *in planta* is unknown, as processing of the fungal amilRNA precursor by plant DCLs may not be favored. However, a lack of DCL1 processing would allow for the systemic movement of the unprocessed precursor *in planta* to sites of infection for its subsequent processing by the protein machinery of the milRNA pathway of the targeted pathogen. Plant transgenes expressing amiRNAs have been commonly utilized in HIGS strategies to induce larval mortality in pest insects [208,209]. More rarely, insect amiRNA backbones have also been investigated. In one pioneering study, an insect amiRNA backbone (*mse-let-7a*) modified to carry a mature sequence complementary to the Ecdysone receptor (*EcR*) of the cotton bollworm (*Helicoverpa armigera*) induced considerable target knockdown when fed to bollworm larvae, corresponding with a significant increase in larval mortality [210]. More recently, plant-expressed insect pre-amiRNAs (plin-amiRNAs) targeting *acetylcholinesterase 2* (*ACE2*) have been demonstrated to effectively induce target silencing and larval mortality in *Helicoverpa armigera* feeding on *N. benthamiana* plants molecularly modified to express plin-amiRNAs [211]. Given that efficient knockdown of insect RNAi targets from plant-expressed dsRNA has been linked to the accumulation of sufficient intact dsRNA [212], parallels can be drawn with those fungal systems that also require sufficient intact dsRNA uptake for efficient silencing, with dsRNA processing within the fungal cell rather than the plant [133]. Both insect miRNA and fungal milRNA precursor transcripts are structurally distinct from plant miRNA precursors, and in the case of plin-amiRNAs, this corresponds to an almost complete failure of plant DCL1-dependent processing of the introduced plin-amiRNAs. This failure, however, would lead to an elevated abundance of intact dsRNA (i.e., the plin-amiRNA) that would be readily processed by the protein machinery of the miRNA pathway of the insect, a processing outcome that could potentially trigger highly efficient RNAi [211]. In this way, amiRNAs delivered via fungal milRNA precursors could be a potent tool for future systems aiming to maximize the availability of intact precursors within the plant host and ultimately in the fungal pathogen. Furthermore, amiRNA-delivering precursor transcripts are generally shorter when they have adopted their specific stem-loop folding structures compared to most exogenously applied conventional dsRNAs. This would enable their efficient loading onto smaller-sized nanoparticles that could move freely through the plant apoplast and be readily internalized into cells of the plant [163,186] or the pathogen.

## 11. Conclusions and Future Considerations

The mechanistic action of fungal RNAi pathways and RNA-based crop protection strategies as an alternative to conventional fungicides will continue to be research areas of significant agricultural importance in the coming years. While RNA-based crop protection can be achieved for some crop species via HIGS, it involves the time-consuming and expensive development of new GM crop varieties, and only in crop species where genetic modification is plausible. On the other hand, RNA-based SIGS represents a non-GM approach for application to all crop species, with the additional potential benefit of being highly responsive in controlling new strains of pathogens as they arise. The signs are very promising for using a SIGS approach to control fungal pathogens that infect the aerial surfaces of plants (e.g., rust fungi and fungal pathogens that spoil fruit and vegetable quality). However, a major challenge facing the SIGS approach is the delivery of the RNA biopesticide to other sites of pathogen infection, particularly in the roots. Unlike conventional fungicides, the sequence-specific nature of both the HIGS and SIGS strategies means that carefully designed RNA constructs will be highly specific in targeting a particular fungal pathogen and minimize off-target effects on other organisms in the environment.

The challenge of developing systemic RNA pesticides is reflected in the efficacy of conventional fungicide sprays in controlling fungal pathogens that infect the roots or internal tissues of plant shoots. While systemic fungicides exist, they are generally limited in their movement and application, and there is still no phloem-mobile foliar fungicide able to provide acceptable control against root-borne fungal pathogens [213]. Indeed, highly systemic fungicides that can enter the xylem and phloem for translocation throughout the plant are rare, with the toxic organophosphate Fosetyl being a notable exception [213]. Furthermore, the physiochemical characteristics that favor the mobility of conventional fungicides in plants may not favor their subsequent uptake into fungal cells, and attempts to chemically alter existing systemic fungicides have yet to yield improved commercial products. Moreover, fungicides are becoming increasingly ineffective as resistance arises in pathogens. These issues associated with the use of traditional fungicides emphasize the potential of the use of exogenous RNA, but only if delivery of the biopesticide can be achieved to all sites of fungal pathogen infection in the plant.

Here, we have outlined what is known about both plant and fungus gene silencing pathways. In particular, we have highlighted some considerable gaps in our current understanding of gene silencing pathways in fungal pathogens. As a further example to illustrate this point, based on the comprehensive knowledge of gene silencing pathways in other organisms, including plants, it is generally assumed that HIGS and SIGS act via a post-transcriptional RNAi mechanism, but exogenous RNA-induced transcriptional gene silencing (TGS) of target genes has not been ruled out as a possible mechanism in fungal pathogens. The mechanism of exogenous RNA transport and processing in plants and the selective exchange of regulatory RNA between plants and their diverse fungal pathogens also need further research. Moreover, while there are large differences between fungal pathogen species regarding the effectiveness of SIGS, the reasons behind these disparities remain largely unknown. Closing these knowledge gaps should enable the development of more effective RNA biopesticides for crop protection against a wide range of fungal pathogens.

There is good reason to be optimistic that RNA pesticides could be engineered to be more effectively transported throughout the plant apoplast and/or symplast and into a fungal pathogen to execute their action. Consider the potential of an advanced RNA biopesticide that is efficiently taken up, stable, and highly mobile in plants and traffics to sites of infection in the plant to control a fungal pathogen. However, achievement of this goal for root-borne fungal pathogens will undoubtedly depend on our ability to gain a greater understanding of the uptake, trafficking, and processing of exogenous RNA in both the plant host and the pathogen in the future.

## Figures and Tables

**Figure 1 ijms-24-12391-f001:**
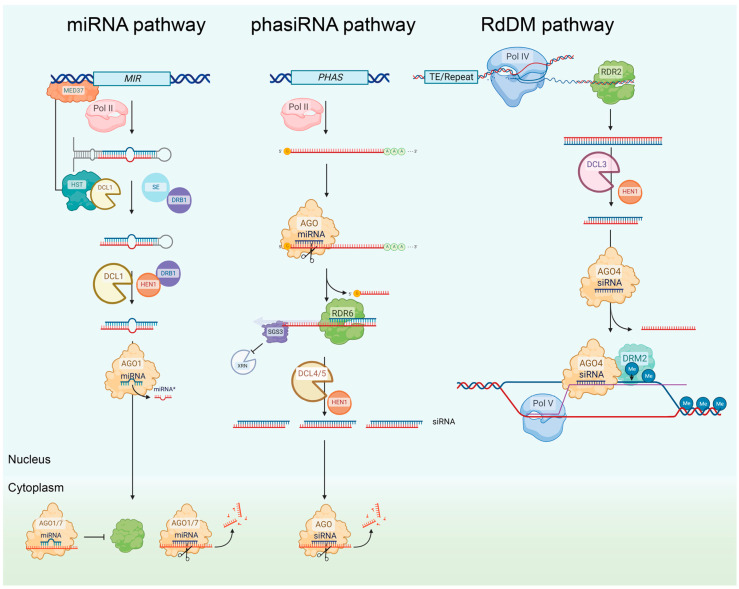
The miRNA, phasiRNA, and rasiRNA (RdDM) pathways in plants. The miRNA pathway (**left**): Endogenous *MIR* loci are transcribed by Pol II to produce pri-miRNAs. The EXPO5 ortholog HST associates with MED37 and DCL1 at *MIR* loci to link pri-miRNA transcription and processing by DCL1, in coordination with SE and DRB1, to produce pre-miRNA transcripts. DCL1/SE/DRB1 processes the pre-miRNA into the miRNA/miRNA* duplex, with each duplex strand methylated at its 3′ end by HEN1 in complex with DRB1 prior to AGO1-mediated export into the cytoplasm. In the cytoplasm, miRNAs guide AGO1 to complementary mRNAs for target cleavage and/or translational inhibition. The phasiRNA pathway (**middle**): endogenous *PHAS* loci are transcribed by Pol II to produce long ssRNA transcripts with a 5′ m^7^G cap and a 3′ poly(A) tail. *PHAS* transcripts are cleaved by 22-nt miRNA-guided AGO1 to remove the 5′ portion of the transcript. The 3′ end of the transcript is converted into a dsRNA by RDR6 and SGS3, which then forms a substrate for DCL4 processing to produce 21-nt tasiRNA in an iterative “phased” manner. These phasiRNAs are exported to the cytoplasm to guide target transcript cleavage by RISC. The biogenesis of tasiRNAs is similar but can also result from the “two-hit” trigger model by AGO7 using a 21-nt miRNA guide to direct the cleavage of non-coding *TAS3* transcripts. The rasiRNA (RdDM) pathway (**right**): Pol IV transcribes non-coding RNAs (ncRNAs) from transposons or other repetitive DNA elements, and such transcripts are co-transcriptionally converted into dsRNA by RDR2. The dsRNA is processed by DCL3 into 24-nt siRNA duplexes, which are subsequently methylated by HEN1 at the 3′ end of each duplex strand. Such duplexes are primarily loaded into AGO4, and the passenger strands are discarded while the guide strands are retained. When scaffold RNA produced by Pol V is recognized by the 24-nt siRNA-AGO4 complex, DRM2 is activated to methylate the cytosine residues of complementary DNA molecules for their silencing at the transcriptional level via RdDM. Figure 1 was created with BioRender.com.

**Figure 2 ijms-24-12391-f002:**
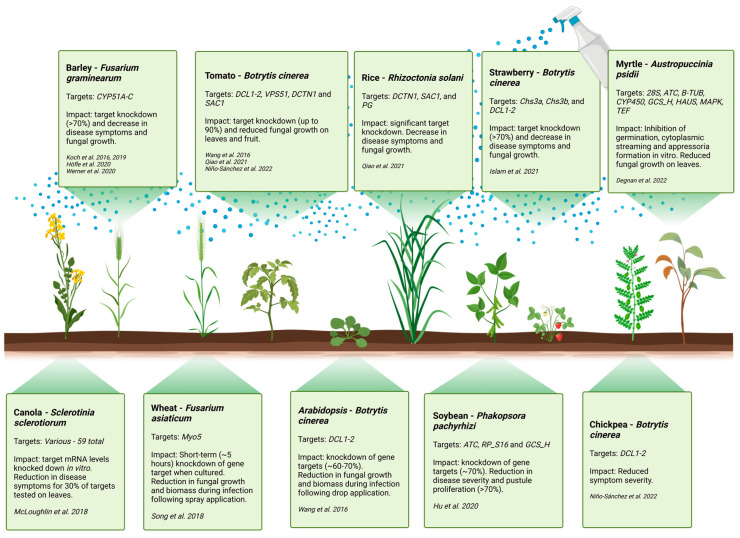
Examples of the control of fungal pathogens by SIGS in plants. The figure was generated based on the experimental findings reported in [16,79,80,102,111,130,131,132,133,135,136,137]. Targets: *CYP51A-C: Cytochrome P450 lanosterol C-14 α-demethylases A-C*; *DCL1-2: DICER-like 1*, *DICER-like 2*; Vesicle trafficking genes: *vacuolar protein sorting 51* (*VPS51*), *dynactin* (*DCTN1*), and *suppressor of actin* (*SAC1*); *Polygalacturonase* (*PG*); *Chs3a*, *Chs3b*: *chitin synthase class III A*, *B*; *Myo5*: *Myosin 5*; *ATC*: *acetyl-CoA acyltransferase*; *RP_S16*: *40S ribosomal protein S16*; *GCS_H*: *glycine cleavage system H protein*; *28S-1: ribosomal RNA 28S-1*; *MAPK*: *mitogen activated protein kinase*; *HAUS*: *uncharacterized haustoria target genes*, *H01215*, *H01136* and *H12890*. Figure 2 was created with BioRender.com.

**Figure 3 ijms-24-12391-f003:**
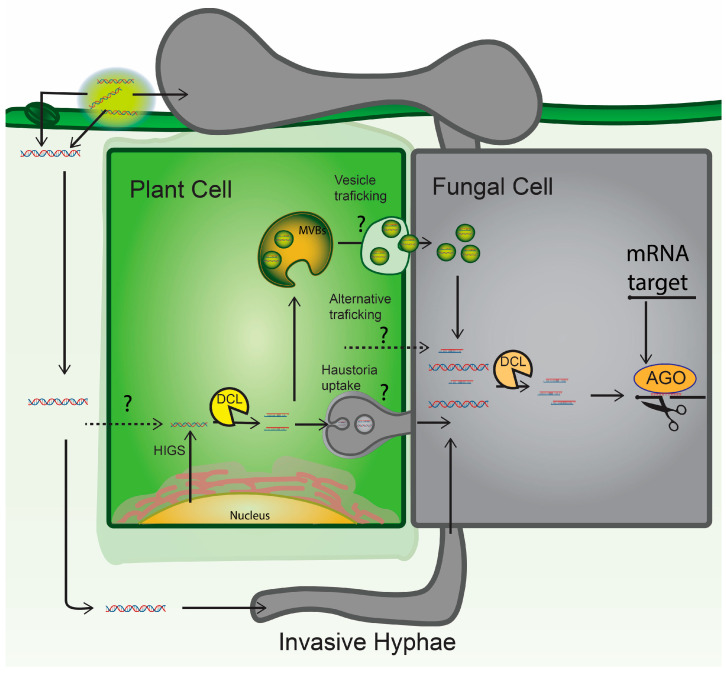
Potential pathways for the uptake of exogenous RNA and SIGS in fungal pathogens. Topically applied dsRNAs can be taken directly into the fungal cells on the plant surface or by first passing through the cells of the plant. In the latter case, the dsRNA could be translocated through the plant apoplast before delivery to the fungal infection hyphae. Alternatively, the exogenous dsRNA may be internalized within the plant cells for processing into sRNAs and thereafter mirror the mechanistic action of HIGS: processed and/or unprocessed dsRNA are taken into the fungal cell either by haustorial feeding, endocytosis, and vesicle trafficking, or via another unknown mechanism. In the fungal cell, the fungal RNAi machinery is guided by siRNAs derived from the exogenous dsRNA to mediate target gene silencing in the pathogen. Alternate, and still to be determined routes of entry of dsRNA/sRNA into the fungal cell are represented by question marks (?).

## Data Availability

Not applicable.

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
