# Peer review of "RNA-Based Control of Fungal Pathogens in Plants"

_ijms, 2023, doi:10.3390/ijms241512391_

Round 1

Reviewer 1 Report

This manuscript is well written and will be available as an introduction to this study area, for college students as well as for researchers. For the publication, minor to moderate revision is required. Four major issues are pointed out first, then followed by minor ones.

Major issues

Spelling

Although English is almost perfect, British and American spellings both are seen: for example favour and favor. Either styling should be used.

Abbreviation

As in other modern scientific reports, a lot of abbreviations appear in the text. A list of abbreviations may help avoid confusion.

Section

The manuscript is consisted of 11 sections. After the third section, no title of the section is shown and also no explanations on the section are given. What is targeted should be explained briefly in the beginning of each section.

Title for figure

No title is given to any figure. Each figure must be shown with its title first and then the legend should be written.

Minor issues

L51-83

An illustration will greatly help readers understand the mechanisms in the text.

L110-112

Refer to citation.

L125-127

Refer to citation. The model plant, Arabidopsis thaliana, is mentioned in the manuscript. However, the authors refer to this plant only with the genus name. This must be avoided, because it cannot be correctly understand what congeneric species are mentioned. The authors “must” write this species in its full scientific name and thereafter refer to this plant as A. thaliana.

Figure 1

RdDM pathway is not mentioned in the text. It is written as tasiRNA pathway. The same term should be used both in the text and the figure.

L298

The species, Puccinia graminis f. sp. tritici is abbreviated as “Pgt” in the text. The abbreviation should not be italicised.

L294

The abbreviation for day post-infection should be considered anyone other than dpi, which is widely used for dot-per-inch.

L649

What is the genus name of A. psidii? Does it belong to Arabidopsis? Here, the genus should be spelled out in order to avoid from confusing with this plant.

L706

If only one species is mentioned here, refer to the complete species name. If two or more congeners are considered, it should be written as Puccinia spp.

Figure 3

The figure legend should be placed just below the figure. Do not insert text between the figure and its legend.

L1230-1232

Refer to citation.

No serious problems in English were found in the manuscript. With the correction by the authors, a revised manuscript can be accepted for the publication in the journal. I do not find any need to review the revised manuscript.

Author Response

RNA-based control of fungal pathogens in plants (MS ID: ijms-2501657)

The authors thank the 3 reviewers for their positive and constructive assessment of our review article, and below, we outline the modifications we have made to our article to address each point raised by Reviewers 1 to 3.

Please note the variation between the line numbers of the .docx and .pdf files. Below, and in all of the reviewer suggestions, the .pdf line numbers are used unless otherwise specified below.

Kind regards,

Andrew Eamens (on behalf of all authors)

Reviewer #1

Comments and Suggestions for Authors

This manuscript is well written and will be available as an introduction to this study area, for college students as well as for researchers. For the publication, minor to moderate revision is required. Four major issues are pointed out first, then followed by minor ones.

The authors thank Reviewer #1 for their detailed assessment of our Review article, and we have addressed each comment raised by Reviewer #1 as outlined below.

Major issues

Spelling

Although English is almost perfect, British and American spellings both are seen: for example favour and favor. Either styling should be used.

The authors thank Reviewer #1 for identifying this inconsistency. We have changed the spelling of the highlighted words on lines 1213, 1158, 1118, 1087 and 501.  We also detected one further spelling consistency issue, and have changed it accordingly: line 420.  

Abbreviation

As in other modern scientific reports, a lot of abbreviations appear in the text. A list of abbreviations may help avoid confusion.

The authors thanks Reviewer #1 for this suggested improvement to our Review article. We have generated a List of Abbreviations in the revised version of our article. The list has been placed after the Keywords section of the Review (page 1).

Section

The manuscript is consisted of 11 sections. After the third section, no title of the section is shown and also no explanations on the section are given. What is targeted should be explained briefly in the beginning of each section.

The authors respectfully disagree with this comment raised by Reviewer #1. Namely, each of the eleven sections of the Review article already has a title, with several of the sections further broken down into subsections. Therefore, the authors are of the opinion that the existing format of our Review article provides a topic sentence, or introductory paragraph, to introduce and/or frame the text of each section / subsection.

Title for figure

No title is given to any figure. Each figure must be shown with its title first and then the legend should be written.

In authoring the original version of the Review article, the authors followed the formatting convention of the journal as required. Therefore, the first sentence of each Figure legend forms the title of each Figure. In the revised version of our article, we have ‘bolded’ the text of each Figure title to assist the reader. However, the authors understand that this formatting change may be converted back to its original format by the journal. 

Minor issues

L51-83

An illustration will greatly help readers understand the mechanisms in the text.

The authors agree with this Reviewer #1 comment. Figures 1 to 3 were included in the original version of our article to assist reader understanding of the mechanistic basis of the RNA silencing pathways and RNAi technologies discussed in the text. The authors are therefore of the opinion that placement of the Figures following the text which describes the information presented in each Figure forms an appropriate presentation style.

L110-112

Refer to citation.

Citation has been added to the revised article.

L125-127

Refer to citation.

These lines are to introduce the subsequent sections, which are heavily cited. Wording was clarified to explain this.

The model plant, Arabidopsis thaliana, is mentioned in the manuscript. However, the authors refer to this plant only with the genus name. This must be avoided, because it cannot be correctly understand what congeneric species are mentioned. The authors “must” write this species in its full scientific name and thereafter refer to this plant as A. thaliana.

It is very common for Arabidopsis thaliana to be written without the species name in this field, and the authors are in the practice of following this convention (see, for example, that of the sixteen references with Arabidopsis in the title only one includes the full species name). While we appreciate the importance given to scientific accuracy, we do not believe this abbreviation to be confusing in this context.

Figure 1

RdDM pathway is not mentioned in the text. It is written as tasiRNA pathway. The same term should be used both in the text and the figure.

The RdDM pathway and rasiRNA pathway are used interchangeably in section 2.1.5. The interchangeable use of this terminology is common practice in this field.

 L298

The species, Puccinia graminis f. sp. tritici is abbreviated as “Pgt” in the text. The abbreviation should not be italicised.

The cited reference italicises “Pgt” and the authors want to ensure that readers are not confused if they were to also read the cited publication.

L294

The abbreviation for day post-infection should be considered anyone other than dpi, which is widely used for dot-per-inch.

The authors would like to maintain the original abbreviations used in the cited work. With the inclusion of the list of abbreviations, any confusion should now be resolved in the revised article.

L649

What is the genus name of A. psidii? Does it belong to Arabidopsis? Here, the genus should be spelled out in order to avoid from confusing with this plant.

To avoid confusion to the reader, the full name of Austropuccinia psidii has been provided. We thank Reviewer #1 for this helpful suggestion

L706

If only one species is mentioned here, refer to the complete species name. If two or more congeners are considered, it should be written as Puccinia spp.

Thank you to Reviewer #1 for identifying this oversight. We have amended the text accordingly.

 Figure 3

The figure legend should be placed just below the figure. Do not insert text between the figure and its legend.

Thank you to Reviewer #1 for identifying this oversight. We have corrected this formatting error in the revised version of our article.

 L1230-1232

Refer to citation.

Thank you identifying this oversight. Citation has been added to the revised article.

Comments on the Quality of English Language

No serious problems in English were found in the manuscript. With the correction by the authors, a revised manuscript can be accepted for the publication in the journal. I do not find any need to review the revised manuscript.

Again, the authors would like to take this opportunity to thank Reviewer #1 for their helpful and constructive suggestions to improve the standard of our originally submitted Review article. The authors are of the opinion that the Reviewer #1 comments that we have addressed in the revised article have improved the quality of the article.

Reviewer 2 Report

My comments can be found in the attached MS.

Quality of English is fine, I see a couple of typos and they were highlighted.

Author Response

RNA-based control of fungal pathogens in plants (MS ID: ijms-2501657)

The authors thank the 3 reviewers for their positive and constructive assessment of our review article, and below, we outline the modifications we have made to our article to address each point raised by Reviewers 1 to 3.

Please note the variation between the line numbers of the .docx and .pdf files. Below, and in all of the reviewer suggestions, the .pdf line numbers are used unless otherwise specified below.

Kind regards,

Andrew Eamens (on behalf of all authors)

Reviewer #2

The authors thank Reviewer #2 for their detailed assessment of our Review article, and we outline below how we have addressed each comment raised by Reviewer #2.

Please note: Reviewer #2 supplied their comments on the manuscript file itself, therefore, our revisions are listed below via providing the line number which corresponds to each comment.

Line 11: Changed to “expanding population”

Line 36: The word “hungry” has been retained to maintain its intended use in the Review article.

Line 42: The word “foodstuff” has been retained as it more accurately encompasses both human and animal feed.

Line 77: References have been added in the revised version of the Review as requested.

Line 81: While nomenclature for RNA-dependent RNA polymerases is not always consistent in the literature, we maintain that RDR should be used for plant RDRs, and RdRp for viral RDRs. (see Wassenegger & Krczal, 2006)

Line 100: Reference has been added in the revised version of the Review as requested.

Line 128: These lines are to introduce the subsequent sections, which are heavily cited. Wording was clarified to explain this. Thank you to Reviewer #2 for this suggestion.

Line 261: We have removed the text italics as suggested by Reviewer #2.

Line 302: Reviewer #2 requested a reference for the following: “The absence of stable transformation systems for Pgt, and for other closely related fungal species, makes elucidation of the finer molecular details of such a pathway more challenging than it has been for plants.” While efforts were made to comply, it is difficult to find suitable references to demonstrate an absence of experimental options in the case of these fungi. In the case of plants, we trust that the detail provided in the preceding section (2.1) should prove adequate proof to the reader that advances in this research area have been made.

Line 481: Reference has been added in the revised version of the Review as requested.

Line 587: Full form included in the revised version of the Review requested.

Line 596: Text italics has been removed in the revised manuscript as suggested by Reviewer #2.

Line 734: Spelling has been corrected. Thank you for identifying this oversight.

Line 937: in this case, P. infestans is mentioned specifically because it is an oomycete not a fungus.

Line 954: Correction amended as requested by Reviewer #2.

Line 1038: Several lines of new text were added to the end of section 8.1 to address the comments of Reviewer #2 regarding regulation, off-target effects, and public safety concerns. This includes the addition of new references.

Reviewer 3 Report

This review paper deals with a topic of emerging relevance in plant protection. It is complete, well organized and very well written, and I recommend its acceptance after minor revision concerning the following adjustments.

Caption of Fig. 1 is too long and should be made more essential; main sections from 3.1 on must be re-numbered as 3., 4., etc.; line 227: correct to 'involved in'; lines 261 and 596: 'spp.', not in italics; lines 289, 446, 702, 707, 734 and 1166: 'f.sp.' not in italics; line 284: correct to 'fungus'; line 382: correct to 'have'; line 415: use semicolon to separate sentences ('...[98]; furthermore,...'); line 427: use semicolon; lines 446, 702, 734 and 1166: use the abbreviated form 'P. striiformis'; line 525: delete 'in'; line 535: 'Arabidopsis' in italics; line 542: use the abbreviated form 'V. mali'; line 592: use bracketed number of this reference instead of year (2021); the same at line 771. Bracketed numbers should also be used for references considered in Fig. 2? line 596: delete 'other filamentous pathogens, including'; line 646: correct to 'fumigatus'; line 684: use semicolon after 'cinerea'; likewise, at line 915 use semicolon after 'spaces'; line 706: use the abbreviated form 'P. graminis'; line 921: use the abbreviated form 'Z. tritici'; line 952: 'of' instead of 'by'; lines 1116-1117: virus names not in italics; line 1166: delete '(wheat stripe rust)', as this pathogen was previously repeatedly mentioned.

Author Response

Response to reviewers’ comments

RNA-based control of fungal pathogens in plants (MS ID: ijms-2501657)

The authors thank the 3 reviewers for their positive and constructive assessment of our review article, and below, we outline the modifications we have made to our article to address each point raised by Reviewers 1 to 3.

Please note the variation between the line numbers of the .docx and .pdf files. Below, and in all of the reviewer suggestions, the .pdf line numbers are used unless otherwise specified below.

Kind regards,

Andrew Eamens (on behalf of all authors)

Reviewer #3

The authors would like to thank Reviewer #3 for their careful appraisal of the submitted manuscript. We have addressed each of the Reviewer #3 comments in the revised manuscript version as outlined below:

Caption of Fig. 1 is too long and should be made more essential:

The authors acknowledge the length of the legend of Figure 1. However, the degree of textual explanation is required to adequately outline the complexity of the RNA silencing pathways involved. Although we have not reduced the length of the Figure 1 legend in the revised manuscript version, we have made text formatting changes in the attempt to make the text more readily digestible.

main sections from 3.1 on must be re-numbered as 3., 4., etc.:

The authors thank Reviewer #3 for making this suggestion. However, the authors are of the opinion that the current approach to sub-sectioning the text is appropriate for this style of Review article.

Minor corections

We thank Reviewer #3 for the careful review of the text of our article. This was extremely helpful in the revision process. Please see below for specific changes.

line 227: correct to 'involved in';

Thank you for identifying this oversight – text has been corrected in the revised manuscript.

 lines 261 and 596: 'spp.', not in italics;

Thank you for identifying this oversight – text has been corrected in the revised manuscript.

 lines 289, 446, 702, 707, 734 and 1166: 'f.sp.' not in italics;

Thank you for identifying this oversight – text has been corrected in the revised manuscript.

 line 284: correct to 'fungus';

Thank you for identifying this oversight – text has been corrected in the revised manuscript.

 line 382: correct to 'have';

Thank you for identifying this oversight – text has been corrected in the revised manuscript.

 line 415: use semicolon to separate sentences ('...[98]; furthermore,...');

Thank you for identifying this oversight – text has been corrected in the revised manuscript.

 line 427: use semicolon;

Corrected.

lines 446, 702, 734 and 1166: use the abbreviated form 'P. striiformis';

Thank you for identifying this oversight – text has been corrected in the revised manuscript at line numbers 446, 702, 1166. However, on line 734, the abbreviation “Pst” was introduced as it was deemed important due to the milRNA nomenclature in this section and the cited reference. Therefore, no change to the existing text was made.

 line 525: delete 'in';

Thank you for identifying this oversight – text has been corrected in the revised manuscript.

 line 535: 'Arabidopsis' in italics;

Thank you for identifying this oversight – text has been corrected in the revised manuscript.

 line 542: use the abbreviated form 'V. mali';

Thank you for identifying this oversight – text has been corrected in the revised manuscript.

 line 592: use bracketed number of this reference instead of year (2021);

Bracketed numbers have been added at the end of the sentence, but we have also kept the publication dates when authors are named directly as per standard citation practice.

 the same at line 771. Bracketed numbers should also be used for references considered in Fig. 2?

Bracketed numbers for Figure 2 are in the text preceding the figure, and directly relating to Figure 2.

line 596: delete 'other filamentous pathogens, including';

Thank you for identifying this oversight – text has been corrected (and surrounding text was also modified) in the revised manuscript. 

line 646: correct to 'fumigatus';

Thank you for identifying this oversight – text has been corrected in the revised manuscript.

 line 684: use semicolon after 'cinerea'; likewise, at line 915 use semicolon after 'spaces';

Thank you for identifying this oversight – text has been corrected in the revised manuscript.

 line 706: use the abbreviated form 'P. graminis';

We have retained the use of the full species name at this point in the revised Review version due to the large gap since its previous mention.

 line 921: use the abbreviated form 'Z. tritici';

We have retained the use of the full species name at this point in the revised review version due to the large gap since its previous mention.

 line 952: 'of' instead of 'by';

Wording changed as requested.

 lines 1116-1117: virus names not in italics;

All virus names should be italics, therefore, we have not made this suggested correction.

line 1166: delete '(wheat stripe rust)', as this pathogen was previously repeatedly mentioned

Correction has been made in the revised manuscript version.